# Impact of the gut microbiota on the m$^6$A epitranscriptome of mouse cecum and liver

Sabrina Jabs [1✉], Anne Biton[2,5], Christophe Bécavin [2,5], Marie-Anne Nahori[1], Amine Ghozlane [2], Alessandro Pagliuso[1], Giulia Spanò[1], Vincent Guérineau[3], David Touboul [3], Quentin Giai Gianetto[2,4], Thibault Chaze [4], Mariette Matondo [4], Marie-Agnès Dillies[2] & Pascale Cossart[1✉]

The intestinal microbiota modulates host physiology and gene expression via mechanisms that are not fully understood. Here we examine whether host epitranscriptomic marks are affected by the gut microbiota. We use methylated RNA-immunoprecipitation and sequencing (MeRIP-seq) to identify N6-methyladenosine (m$^6$A) modifications in mRNA of mice carrying conventional, modified, or no microbiota. We find that variations in the gut microbiota correlate with m$^6$A modifications in the cecum, and to a lesser extent in the liver, affecting pathways related to metabolism, inflammation and antimicrobial responses. We analyze expression levels of several known writer and eraser enzymes, and find that the methyltransferase Mettl16 is downregulated in absence of a microbiota, and one of its target mRNAs, encoding S-adenosylmethionine synthase Mat2a, is less methylated. We furthermore show that *Akkermansia muciniphila* and *Lactobacillus plantarum* affect specific m$^6$A modifications in mono-associated mice. Our results highlight epitranscriptomic modifications as an additional level of interaction between commensal bacteria and their host.

[1] Unité des Interactions Bactéries-Cellules, Institut Pasteur, U604 Institut National de la Santé et de la Recherche Médicale, USC 2020 Institut National de la Recherche Agronomique, 25 rue du Dr Roux, F-75015 Paris, France. [2] Hub de Bioinformatique et Biostatistique – Département Biologie Computationnelle, Institut Pasteur, USR 3756 CNRS, 28 rue du Dr Roux, F-75015 Paris, France. [3] Institut de Chimie des Substances Naturelles, CNRS UPR 2301, Université Paris-Sud, Université Paris-Saclay, 91198 Gif-sur-Yvette, France. [4] Unité de spectrométrie de masse et Protéomique, CNRS USR 2000, Institut Pasteur, 28 rue du Dr Roux, F-75015 Paris, France. [5] These authors contributed equally: Anne Biton, Christophe Bécavin. ✉email: sabrina.jabs@pasteur.fr; pcossart@pasteur.fr

Posttranscriptional mRNA modifications, most notably m6A[1,2], have recently been shown to contribute to the regulation of mRNA fate by affecting mRNA stability, splicing events or the initiation of translation[3]. mRNA can be methylated by RNA-methyltransferases in specific positions that are mainly located at the 3′ untranslated regions (UTR) and the coding sequence (CDS) of the transcript, utilizing S-adenosylmethionine (SAM) as a methyl donor. Methyl-transferase like (Mettl) 3 in complex with Mettl14 is the most important m6A-modifying enzyme[4] ('writer'), but for specific transcripts, Mettl16 has been proposed to act as an additional N6-adenosine-methyltransferase[5–7]. The demethylases Alkbh5 and Fto ('erasers') can remove m6A modifications[8]. Mutations in Fto have been shown to be associated with obesity in humans and mice[9–11], identifying Fto as an important co-regulator of host metabolism. m6A modification of mRNA is important in embryonic stem cell and immune cell differentiation[12–14], neurogenesis and neuronal function[15,16], stress responses[17], the circadian rhythm[18], and viral infection[19–24]. A less prevalent epitranscriptomic modification induced by the recently identified writer protein Pcif1[25] is m6Am[26]. It is mostly found at the first encoded nucleotide adjacent to the 7-methylguanosine cap and enhances the stability of mRNAs[27]. The commonly used method of mapping m6A modifications, methylated RNA-immunoprecipitation and sequencing (MeRIP-Seq), also detects m6Am modifications along with m6A-modifications. However, m6A is far more abundant than m6Am, and m6Am only occurs at a very specific position in the 5′ cap, therefore the majority of modifications detected by MeRIP-Seq are likely to be m6A.

Commensal bacteria, in particular the gut microbiota, have profound effects on host physiology, including host metabolism, intestinal morphology, the development of the immune system, and even behavior[28]. Gut microbial metabolites and fermentation products, e.g. short chain fatty acids (SCFA), sphingolipids, polyamines, and tryptophan metabolites have been shown to partially mediate the influence of gut commensals on their host by modulating transcription and epigenetic modifications[29–33]. However, a complete understanding of mechanisms underlying gut-microbiota-host interactions still remains elusive. Recent reports have suggested that changes in m6A levels are associated with inflammatory states[14,34] and very recently, the presence of a microbiota has been suggested to induce changes in epitran-scriptomic profiles in the brain, and several other tissues[35].

By using MeRIP-Seq, we set out to determine if the presence of the microbiota as a whole, and of specific commensals is associated with changes in host epitranscriptomic m6A mRNA modification profiles in mouse cecum and liver. We find m6A modification profiles in both tissues to be influenced by the presence of a gut flora. Furthermore, we show that monoassociation of mice with the commensal bacteria *Akkermansia muciniphila* and *Lactobacillus plantarum* influences m6A modification profiles in cecum and liver.

## Results

**Expression of m6A 'writer' and 'eraser' proteins.** As a prerequisite to our analysis of m6A modifications, we analyzed expression levels of the methyltransferases Mettl3, Mettl14, Mettl16, and Pcif1 ('writers') and the demethylases Alkbh5 and Fto, which are known to be ubiquitously expressed[36]. mRNA levels determined by qRT-PCR for Mettl3, Mettl14, the m6Am methyltransferase Pcif1 and the demethylases Fto and Alkbh5 confirmed a comparable expression in liver, brain, and different parts of the intestine (small intestine, cecum, and colon) (Supplementary Fig. 1a). Depending on the housekeeping gene used for the determination of relative

mRNA expression, Hprt or Gapdh, their expression in the spleen was two- to six-fold higher than in liver or cecum on mRNA levels (Supplementary Fig. 1a). Western blotting revealed similar levels of Mettl3, Mettl14, Mettl16, and Alkbh5 in the liver, small intestine, colon, and cecum (Supplementary Fig. 1b). In the brain, expression levels of Mettl3, Mettl14, Mettl16, and Alkbh5 were 3–4-fold higher than in intestinal tissues and the liver. We chose to analyze the epitranscriptome in the intestine, and more precisely in the cecum, which is in close contact with the gut microbiota and undergoes profound physiological and morphological changes in the absence of a microbiota[37]. We further included the liver, whose gene expression is also known to be influenced by commensal bacteria[38]. Although the expression of methyltransferases and demethylases was high in the spleen at mRNA and protein levels (Supplementary Fig. 1a, b), we chose not to focus on this organ to study changes in the m6A epitranscriptome influenced by the microbiota, since it is known that several populations of immune cells are strongly reduced in mice without a gut flora[37], which would complicate a quantitative analysis.

As another prerequisite for our study, we verified that the MeRIP-Seq technique was adequate to perform differential methylation analysis. To this end, we compared the recovery of green fluorescent protein (GFP) transcript in vitro transcribed in the presence of m6A by m6A-immunoprecipitations from different RNA preparations or buffer alone (Supplementary Fig. 2a). We were able to recover equal amounts of the GFP-m6A-transcript from total RNA, ribodepleted RNA, purified mRNA, and immunoprecipitation (IPP), indicating that a differential expression analysis from MeRIP-Seq is possible.

**m6A modification profiles in cecum and liver.** We used a series of mice with different gut microbiota (see below) and analyzed methylation profiles using MeRIP-Seq. Overall, using 64 datasets for cecum and 33 datasets for liver with a median coverage of 8× and 11×, respectively (Supplementary Data 1), we detected 36,935 m6A peaks in the various anti-m6A-immunoprecipitates from murine cecum and 25,808 m6A peaks in the anti-m6A-immunoprecipitates from liver (Supplementary Data 2). Fifty-two percent of m6A peaks detected in the cecum were also found in the liver, and 74% of m6A peaks detected in the liver were also found in the cecum. In total, 80 and 84% of the peaks we detected in the cecum and liver, respectively, are described in the Methyl Transcriptome Data Base (MeT-DB v2.0)[39] (Fig. 1a), indicating that we identified bona fide methylation peaks. m6A peaks across different murine and human tissues and cell lines have been described to be largely conserved[1,2], and we found a strong overlap of the peaks we detected with all the different murine types of tissues and cell lines present in the MeT-DB v2.0 database, but also a substantial overlap with published m6A peaks from human tissues and cell lines (Supplementary Fig. 2b). GUITAR plots summarizing the positions of all the m6A modifications for the cecum (Fig. 1b) showed that m6A peaks were mostly present in the CDS and 3′ UTR of mRNA, and at a lesser extent in the 5′ UTR, in agreement with previous studies[1,2]. We performed motif searches on the list of all detected peaks and assessed presence of known motifs for m6A (RRACH)[1,2] and m6Am (NBCAN)[26] separately for the 5′UTR, CDS and 3′UTR (Fig. 1c). As expected, the RRACH motif was less present in the 5′ UTR of transcripts, where instead the consensus motif for m6Am modification (NBCAN) was enriched (Fig. 1c).

**Gut microbiota controls m6A modifications in mouse cecum.** We first compared m6A marks in the cecal transcripts of conventional (CONV) and germ-free (GF) mice and found 440 m6A peaks on 312 transcripts to be differentially methylated. Multidimensional

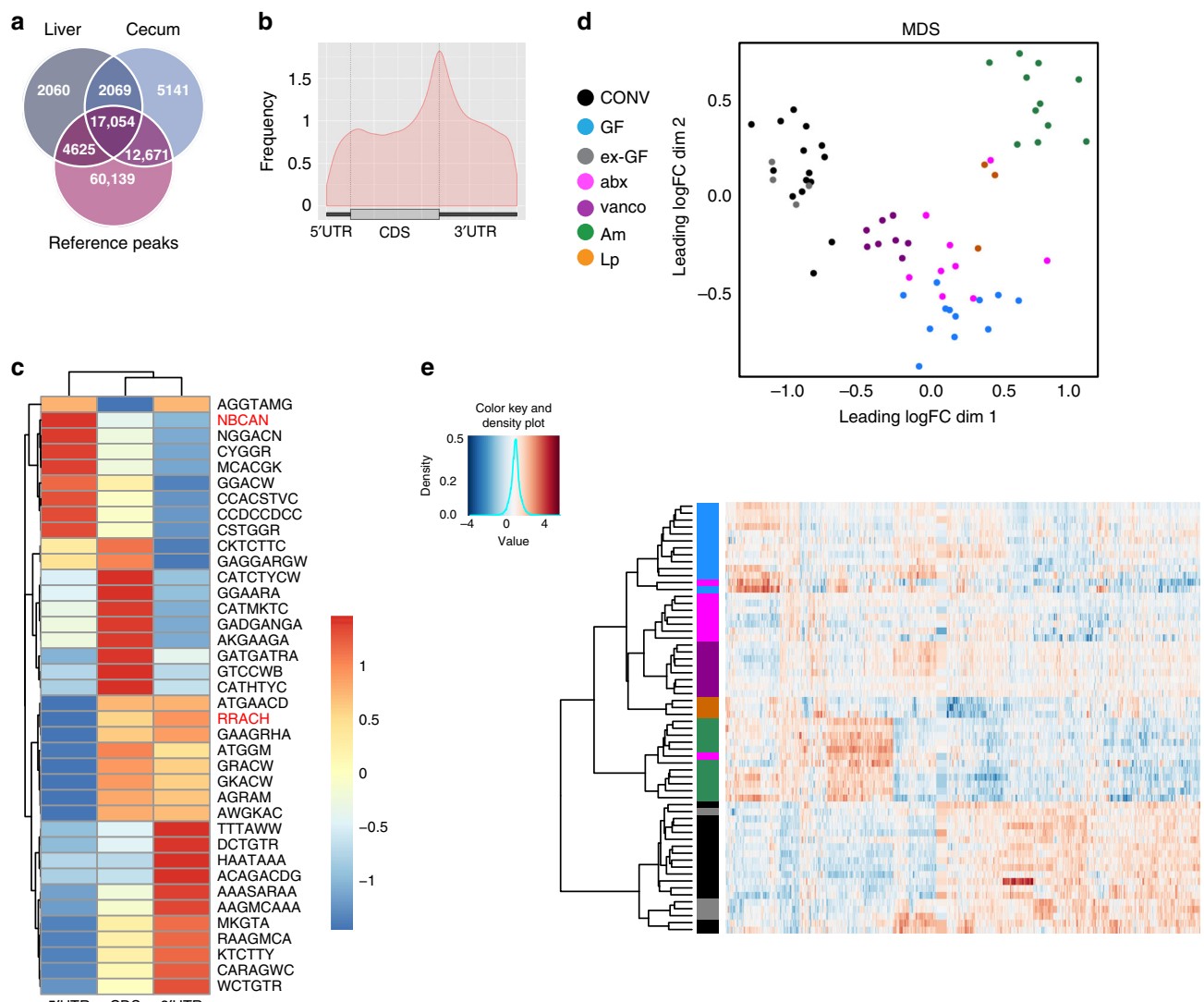

**Fig. 1 Microbiota influences m⁶A modification in mouse cecum. a** 67% of m⁶A peaks in the cecum are overlapping with at least one liver peak, and 87% of detected peaks in the cecum are overlapping a MeT-DB v2.0 reference peak. For the liver tissue, 86% of the methylation peaks are overlapping at least one peak of the cecal tissue, and 89% are overlapping a MeT-DB v2.0 reference peaks. The reference peaks may be overlapping with several peaks we detected. **b** Positions of detected m⁶A peaks on all methylated transcripts in cecum were determined using the GUITAR package. **c** Motif enrichment in m⁶A modification was determined by calculating total occurrence of motifs in m⁶A peaks on the 5′UTR, CDS and 3′UTR of transcripts in the cecum. Consensus motifs for m⁶A (RRACH) and m⁶Am (NBCAN) are highlighted in red. **d** Multi-dimensional scaling (MDS) plot of the peak log2 counts-per-million IP data of all differentially methylated peaks showing the positions of the samples in the space spanned by the first and second MDS dimensions. Samples are colored with respect to condition: CONV (*n* = 15), conventionally raised mouse (black); GF (*n* = 12), germ-free mouse (cyan); ex-GF (*n* = 4), GF mouse colonized with the intestinal content of CONV mice (gray); abx (*n* = 9), CONV mice whose gut microbiota has been depleted by antibiotics treatment (magenta); vanco (*n* = 8), vancomycin/amphotericinB-treated mice (violet); Am (*n* = 11), *A.muciniphila*-mono-colonized mice (green), Lp (*n* = 3), *L.plantarum*-mono-colonized mice (orange); data were obtained from two independent sequencing experiments using ribodepleted RNA or purified mRNA from murine cecum. **e** Heat map of the peak log2 counts-per-million IP data of all differentially methylated peaks. Hierarchical clustering was performed using euclidean distance and ward.D2 linkage.

Scaling (MDS) analysis of m⁶A peaks revealed a clear separation between methylation peaks of CONV and GF mice in the cecum (Fig. 1d). Furthermore, the heat map with hierarchical clustering of IP samples based on the differentially methylated peaks (Fig. 1e) showed a strong separation of GF and CONV cecal transcripts. To further investigate whether differential methylation was mediated by the gut microbiota, we colonized GF mice with the intestinal content of CONV mice (ex-GF). After 4 weeks, these mice exhibited similar patterns of the most abundant gut bacterial genera (e.g. *Lachnospiraceae*, *Allistipes*, *Bacteroides*, *Prevotellaceae*, *Akkermansia*) as CONV mice (Supplementary Fig. 3a–d). m⁶A marks in the ceca of ex-GF mice clustered with the methylation peaks of

CONV mice in hierarchical clustering and multidimensional scaling (MDS), and no m⁶A peaks were differentially methylated when comparing CONV and ex-GF mice (Fig. 2a, Supplementary Data 2), demonstrating unambiguously that the gut microbiota mediates differential methylation.

To confirm these results, we analyzed mice treated with a mixture of several antibiotics (vancomycin, metronidazole, neomycin, ampicillin) and the antifungal amphotericin B for three weeks (abx mice), which resulted in an efficient depletion of the gut microbiota (Supplementary Fig. 3e) with very few genera that were still detectable by 16S rRNA sequencing (Supplementary Fig. 3a–c). m⁶A-modified marks in transcripts from abx mice ceca clustered

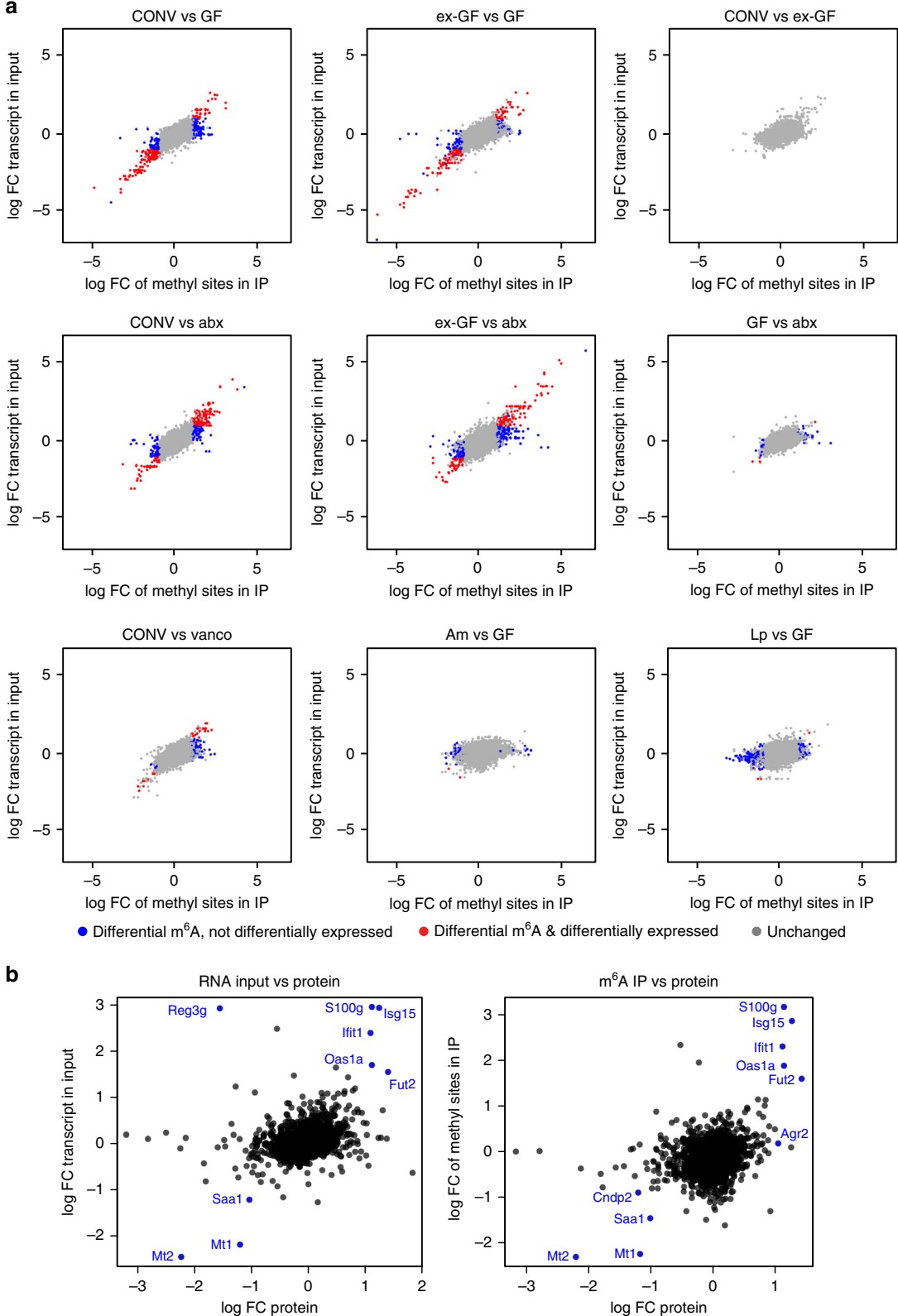

**a**

*CONV vs GF* · *ex-GF vs GF* · *CONV vs ex-GF* · *CONV vs abx* · *ex-GF vs abx* · *GF vs abx* · *CONV vs vanco* · *Am vs GF* · *Lp vs GF*

• Differential m⁶A, not differentially expressed   • Differential m⁶A & differentially expressed   • Unchanged

**b**

*RNA input vs protein* · *m⁶A IP vs protein*

close to those in GF mice (Fig. 1d, e), demonstrating that the impact of a conventional microbiota on m⁶A peaks can be suppressed by antibiotics treatment. A small number of m⁶A peaks (60) was still differentially methylated when comparing abx and GF mice (Fig. 2a, Supplementary Data 2), which might be due to the influence of the few commensals still present after antibiotics treatment (Supplementary Fig. 3a–c, e), or the antibiotic treatment itself.

To gain further insight into the bacterial species that may be involved in the regulation of differential m⁶A modifications, we treated mice with only vancomycin, to which we equally added the antifungal amphotericin B (vanco mice). This treatment resulted in a strong and reproducible enrichment of several genera (*Akkermansia, Escherichia/Shigella, Lactobacillus, Lachnospiraceae, Parasutterella*; Supplementary Fig. 3a–d). In the

**Fig. 2 Differential methylation and protein expression profiles in the cecum. a** $m^6A$-peaks found to be differentially methylated compared to differential expression of the entire transcripts in indicated mice. Differentially methylated peaks that are also differentially expressed on transcript level are displayed in red, differentially methylated peaks that are unchanged on transcript level, in blue. $m^6A$ peaks that were not significantly changed are shown in gray. Cutoffs for differential expression are log fold change (FC) −1 to 1 and Benjamini–Hochberg-corrected $p$-values < 0.05. $p$-values were estimated from moderated t-statistics with empirical Bayes moderation using limma package, followed by Benjamini-Hochberg correction. **b** correlation of proteome, transcriptome and epitranscriptome of CONV and GF mice; log FC of protein levels between CONV and GF mice was compared to log FC of transcript input levels (left panel) or log FC of differentially methylated peaks (right panel) between CONV and GF mice; CONV $n = 7$, GF $n = 6$ for proteome analysis; CONV $n = 15$, GF $n = 12$ for transcriptome and differential methylation analysis. Targets significantly altered between CONV and GF on protein and transcript and $m^6A$ methylation, respectively, are displayed.

MDS analysis, $m^6A$ peaks in vanco mice formed a cluster that was distinct from the GF/abx and CONV/ex-GF mouse clusters (Fig. 1d, e). As expected, vanco mice exhibited less differentially methylated peaks compared to CONV mice (97) than GF (440) or abx (374) mice, respectively, compared to CONV mice (Fig. 2a, Supplementary Data 2). These data suggest that the persisting commensals in vanco mice are involved in the gut microbiota-mediated regulation of $m^6A$ modification patterns observed in CONV cecum.

Importantly, 62% of differentially methylated peaks between GF and CONV, and 83% of all differentially methylated $m^6A$ peaks including all conditions, were not significantly changed in the expression levels of the corresponding entire transcript (Fig. 2a, Supplementary Data 2), demonstrating that differential methylation is not merely a consequence of differences in gene expression between the biological conditions, as it had been previously suggested[40]. Similar to the situation in GF mice compared to CONV mice, differentially methylated transcripts in abx mice, were mostly (52%) unchanged at the transcript level (Fig. 2a).

Since $m^6A$ modifications are linked to both stabilization and destabilization of transcripts[27,41], a complete uncoupling of levels of differential methylation and differential transcript expression, however, would be very surprising. When we examined levels of the residual 38% of peaks that were differentially methylated and at the same time differentially expressed (Supplementary Data 3), increased methylation correlated with an increased abundance of the transcript in most cases. This suggests that a minor proportion of transcripts with $m^6A$ peaks that we found to be influenced by the microbiota are stabilized, as it is also the case in viral infection[42,43]. We then performed label-free proteomics analysis to assess whether differential methylation and transcript levels correlate with differential protein expression levels in the cecum of CONV and GF mice. As expected from previous studies trying to correlate transcriptomic and proteomic data[44], the correlation was weak (Fig. 2b). However, we found several transcripts to be differentially methylated and differentially expressed at protein levels, and in most cases, increased methylation levels correlated with increased transcript levels and increased protein expression (Fig. 2b). The correlation was the strongest in transcripts involved in immune responses and inflammatory diseases, such as Isg15, Oas1a, and Fut2.

$m^6A$-modifications have been reported to influence alternative splicing and differential exon usage[45–47]. We therefore performed a differential transcript isoform usage (DTU) analysis. Applying a cut-off of 5% on $p$-values adjusted both at the gene- and isoform-level, we detected only six genes showing evidence of differential isoform usage between CONV and GF mice, none of which was overlapping with a differentially methylated peak (Supplementary Data 3). Across all biological conditions, only 3% of differentially methylated peaks were present on genes displaying differential isoform usage (Supplementary Data 3), suggesting that differential exon usage is not strongly influenced by microbiota-induced differential methylation.

Taken together, our results establish that the gut microbiota regulates posttranscriptional mRNA modifications in the host in addition to well-known effects on transcription and protein expression.

**Distinct bacterial species influence $m^6A$-modification profiles.** To test if differential methylation can also be induced by mono-association of GF mice with single bacterial species, we chose representatives of two genera (*Akkermansia* and *Lactobacillus*) that were enriched in vancomycin-treated mice (Supplementary Fig. 3a–c): *Akkermansia muciniphila* and *Lactobacillus plantarum* have both been shown to influence host physiology[48,49]. We successfully mono-associated GF mice with *A.muciniphila* (Am) and *L.plantarum* (Lp; Supplementary Fig. 3f). $m^6A$ methylation profiles from mono-associated mice formed clusters distinct from those found in GF and abx mice in MDS analysis (Fig. 1d, e). In hierarchical clustering of IP samples based on differentially methylated peaks Am and Lp even more clearly clustered separately from GF and abx mice, indicating a specific influence of each of these two bacterial species on host methylation profiles (Fig. 1d). The numbers of differentially methylated peaks between GF mice and Am- (115) and Lp- (460) mono-associated mice, respectively, suggested that *L.plantarum* has a stronger effect on host RNA-methylation than *A.muciniphila*. However, it should be taken to account that the number of samples for Lp mice, which was lower than for the other conditions, may result in a higher number of differentially methylated peaks, as it is typically the case for transcriptomics studies[50]. Interestingly, 99 and 98% of differentially methylated transcripts between GF and Lp, and GF and Am mice, respectively, were not changed at the transcript levels, which was a far higher percentage than for all peaks (83%) and peaks differentially methylated between GF and CONV mice (62%).

**Microbiota regulates $m^6A$ in metabolic and inflammatory pathways.** To decipher the cellular functions affected by microbiota-dependent modifications, we performed pathway analyses on the list of all transcripts differentially methylated between CONV and GF cecum. Ingenuity pathway analysis (IPA) for diseases and molecular and cellular function revealed an enrichment of differentially methylated transcripts involved in inflammatory and microbial responses, metabolic and gastrointestinal diseases. Among molecular and cellular functions, lipid, vitamin, carbohydrate, and amino acid metabolism were enriched, as well as posttranslational modifications important for epithelial integrity. (Fig. 3a–c, Supplementary Data 4). These processes have been described to be influenced by the microbiota[37,51–53] and our findings suggest that this influence is in part mediated by modulation of posttranscriptional mRNA modifications. The differentially methylated transcripts linked to lipid metabolism (Fig. 3b), were both hyper- and hypomethylated in CONV and ex-GF mice compared to GF mice, whereas transcripts involved in protein glycosylation and inflammatory

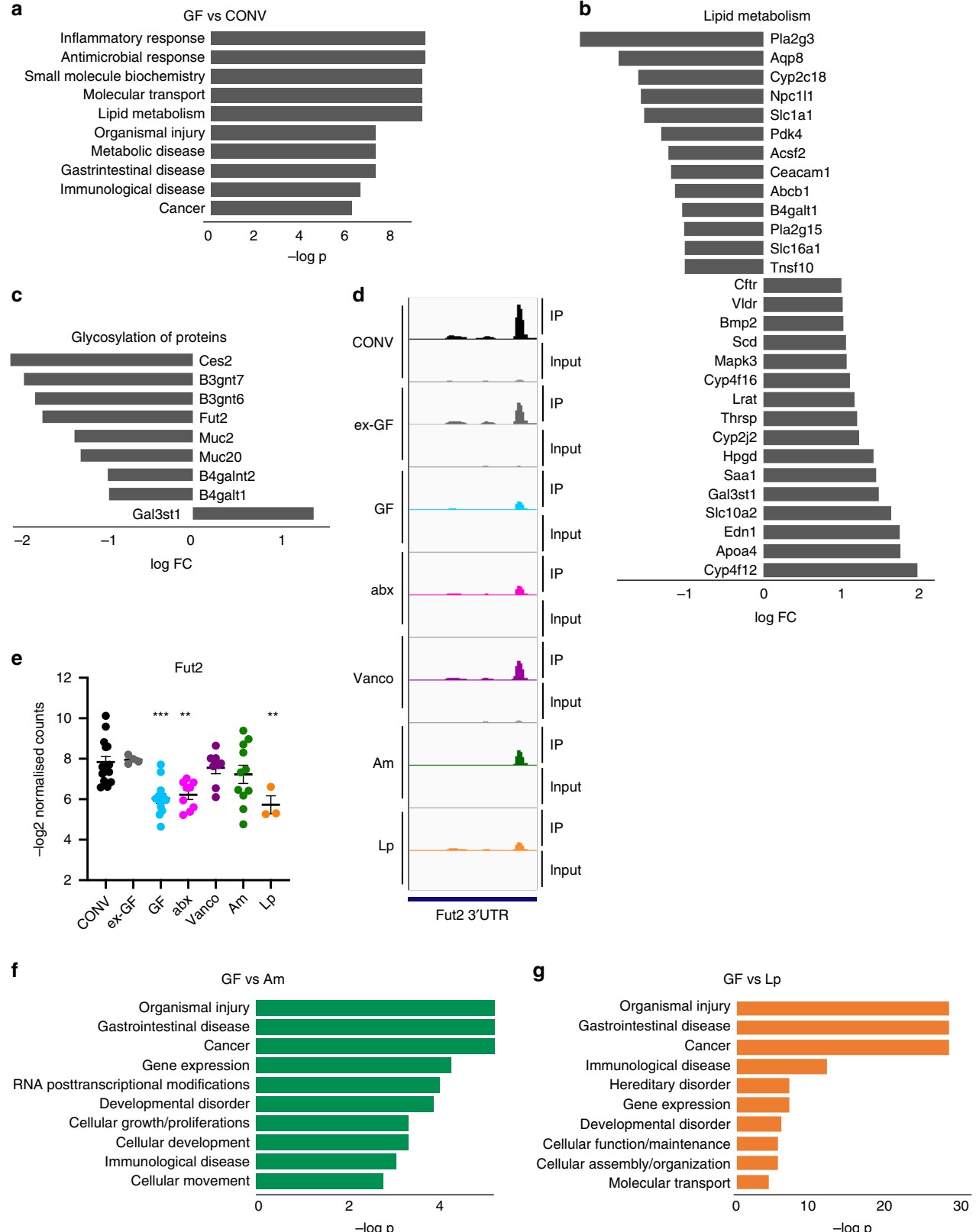

responses, e.g. Fut2, were rather hypomethylated (Fig. 3c–e). Interestingly, distinct peaks on the same transcripts were not necessarily all regulated by the microbiota, suggesting that this influence is peak-specific (Supplementary Data 2). However, we rarely detected hyper- and hypomethylated peaks on the same transcripts. More often differentially methylated peaks occurred

on the same transcript with peaks that were not changed between the different mouse models (Supplementary Data 2).

We next performed IPA pathway analysis for diseases and disorders and molecular and cellular function between GF mice and mice mono-associated with *A.muciniphila* and *L.plantarum*. Diseases strongly influenced by both bacteria were gastrointestinal

**Fig. 3 Differentially methylated transcripts in mouse cecum. a** Pathway analysis of "molecular and cellular function" and "diseases and disorders" of transcripts differentially methylated between GF and CONV mice was performed using Ingenuity pathway analysis (IPA). log -$p$ of the 10 most enriched pathways are displayed. Lists of genes contained in each category and comparison are given in Supplementary Data 4; **b** Expression log ratio of transcripts differentially methylated between GF and CONV mice related to lipid metabolism as defined in IPA. **c** Expression log ratio of transcripts differentially methylated between GF and CONV mice related to glycosylation of proteins as defined in IPA. **d** Representation of Fut2 m$^6$A peak (mean of read per million normalized coverage (RPM) in detected methylation peaks) from anti-m$^6$A immunoprecipitates and input in the 3′UTR of the Fut2 transcript in cecum. Peaks were visualized for the indicated mouse models using IGV; **e** quantification of m$^6$A peak on the fucosyltransferase 2 transcript (Fut2); CONV ($n = 15$), GF, germ-free mouse ($n = 12$); ex-GF ($n = 4$), abx ($n = 9$); vanco ($n = 8$); Am ($n = 11$), Lp ($n = 3$); data were derived from two independent sequencing experiments; data are presented as mean values ±SEM; **$p$-value < 0.01; ***$p$-value < 0.005; $p$-values (all compared to CONV): ex-GF: 0.8402; GF: 0.0002; abx: 0.0019; vanco: 0.7800; Am: 0.3569; Lp: 0.0070; details for statistical analysis are given in the source data file associated to this manuscript. **f, g** Pathway analysis of "molecular and cellular function" and "diseases and disorders" of transcripts differentially methylated between Am and GF mice (**f**), and LP and GF mice (**g**) was performed using Ingenuity pathway analysis (IPA). log -$p$ of the 10 most enriched pathways are displayed. Lists of genes contained in each category and comparison are given in Supplementary Data 4.

diseases and cancer, but also immunological diseases (Fig. 3f, g, Supplementary Data 4). *L.plantarum* induces differential methylation of transcripts involved in cellular function and maintenance, cellular assembly and gene expression, but also gene expression and vitamin metabolism. Importantly, cellular growth and proliferation, and cell death were among the transcripts whose methylation was influenced by *L.plantarum*, (Supplementary Data 4), which correlates well with previously discovered effects on longevity and growth by this bacterium[54,55]. Transcripts differentially methylated between GF and Am mice were equally linked to cell development and gene expression. Interestingly, we found transcripts involved in RNA posttranscriptional modifications to be influenced by *A.muciniphila* (Fig. 3f, Supplementary Data 4).

**Methylation of Mettl16-target Mat2a influenced by the microbiota.** Next, we compared expression levels of Mettl3, Mettl14, Mettl16, Pcif1, Alkbh5, and Fto in CONV and GF tissues. Their mRNA levels were mostly unchanged, except for a significant reduction of Mettl16 mRNA levels in small intestine and colon of GF mice, Mettl3 in small intestine of GF mice, Mettl16 and Alkbh5 in brains of GF mice, and slightly higher levels of Pcif1 mRNA in liver of GF mice (Supplementary Fig. 1c). By Western Blotting, we found Mettl16 slightly, but significantly decreased in the colon of GF mice (Supplementary Fig. 1d) and in the cecum of GF and abx mice compared to the other conditions (Fig. 4a, b), whereas Mettl3, Mettl14, and Alkbh5 were not significantly changed between CONV and GF mice in the cecum (Supplementary Fig. 4a, b). To test whether the changed expression of Mettl16 was the reason for the changes in the epitranscriptome we detected, we compared our differentially methylated transcripts with published targets of METTL16 in human cell lines, identified by knock-down of METTL16 and subsequent MeRIP-Seq[5], or by cross-linking and analysis of cDNA using overexpressed METTL16[6]. Although the overlap between the two published datasets is low, we found a small overlap of our differentially methylated transcripts with both datasets (Supplementary Fig. 5). Interestingly, these transcripts included that for S-adenosylmethionine synthase isoform type-2 (Mat2a), which we found to be hypermethylated in the 3′UTR of CONV, ex-GF and vanco mice (Fig. 4c, d). We could confirm this finding by MeRIP and subsequent quantitative real-time (qRT) PCR (Fig. 4e) and found that the Mat2a protein expression was reduced in GF and abx, cecum compared to CONV, ex-GF and vanco, Am and Lp mice (Fig. 4f, g). Since Mat2a is the enzyme that produces the methyl donor S-adenosylmethionine (SAM) required for methyltransferase activity[56], a reduced Mat2a expression in GF and abx may be linked to the altered methylation profiles.

**Gut microbiota affects m$^6$A modifications in the liver.** To examine whether the influence of the gut microbiota on m$^6$A

mRNA modifications is affecting other tissues than the cecum, we performed differential MeRIP-Seq analysis of transcripts from liver tissue of CONV, GF, Am, and Lp mice, where Mettl14 and Mettl16 expression might be slightly altered in GF mice compared to CONV mice (Supplementary Fig. 4c). We found 527 peaks on 423 transcripts to be differentially methylated across all biological conditions. GUITAR plots revealed the expected peak distribution on the transcripts primarily in the 3′ UTR and along the CDS (Fig. 5a). As for the cecum, we performed motif searches on the list of all detected peaks separately for the 5′UTR, CDS and 3′UTR, and found several motifs to be present, among them the consensus motif for m$^6$A modification (RRACH)[1,2] and the motif for m$^6$Am modification (NBCAN) (Fig. 5b)[26].

The separate clusters of methylation profiles of CONV and GF mice in MDS and hierarchical clustering analyses revealed a clear influence of the microbiota on methylation profiles in the liver (Fig. 5c, d). As in the cecum, the majority (54%) of the 107 differentially methylated peaks when comparing GF and CONV mice, was not found to be differentially expressed at the transcript level (Fig. 5e, Supplementary Data 2). Pathway analyses of Biological Processes and KEGG pathway analysis revealed that, among others, transcripts associated with lipid, vitamin, and amino acid metabolism and insulin signaling were differentially methylated in the liver (Supplementary Data 5).

**Single bacterial species influence m$^6$A modification in the liver.** MDS analysis of GF mice mono-associated with *A.muciniphila and L.plantarum*, revealed profiles of modified m$^6$A-peaks in liver that were clustering with the profiles detected in GF mice (Fig. 5c), suggesting that the association with a single bacterial species only has a small effect on m$^6$A modifications in the liver. However, the heat map with hierarchical clustering based on all differentially methylated peaks for each comparison, showed clearly separated methylation profiles in mice associated with *L.plantarum*, and to a lesser extent with *A.muciniphila*- colonized mice (Fig. 5d). Pathway analysis demonstrated that in GF, Am and Lp mice, respectively, compared to CONV mice, metabolic pathways were the most affected by differential methylation (Supplementary Data 5), some of which have previously been shown to be influenced by the gut microbiota in the liver[51,57,58]. When comparing the methylation profiles in liver of mono-associated mice (Am, Lp) to GF mice, Lp mice displayed a stronger effect on host m$^6$A modifications of the two tested bacterial strains (Supplementary Data 2 and 5), and influenced pathways in insulin signaling, lipid metabolism and cell differentiation (Supplementary Data 5).

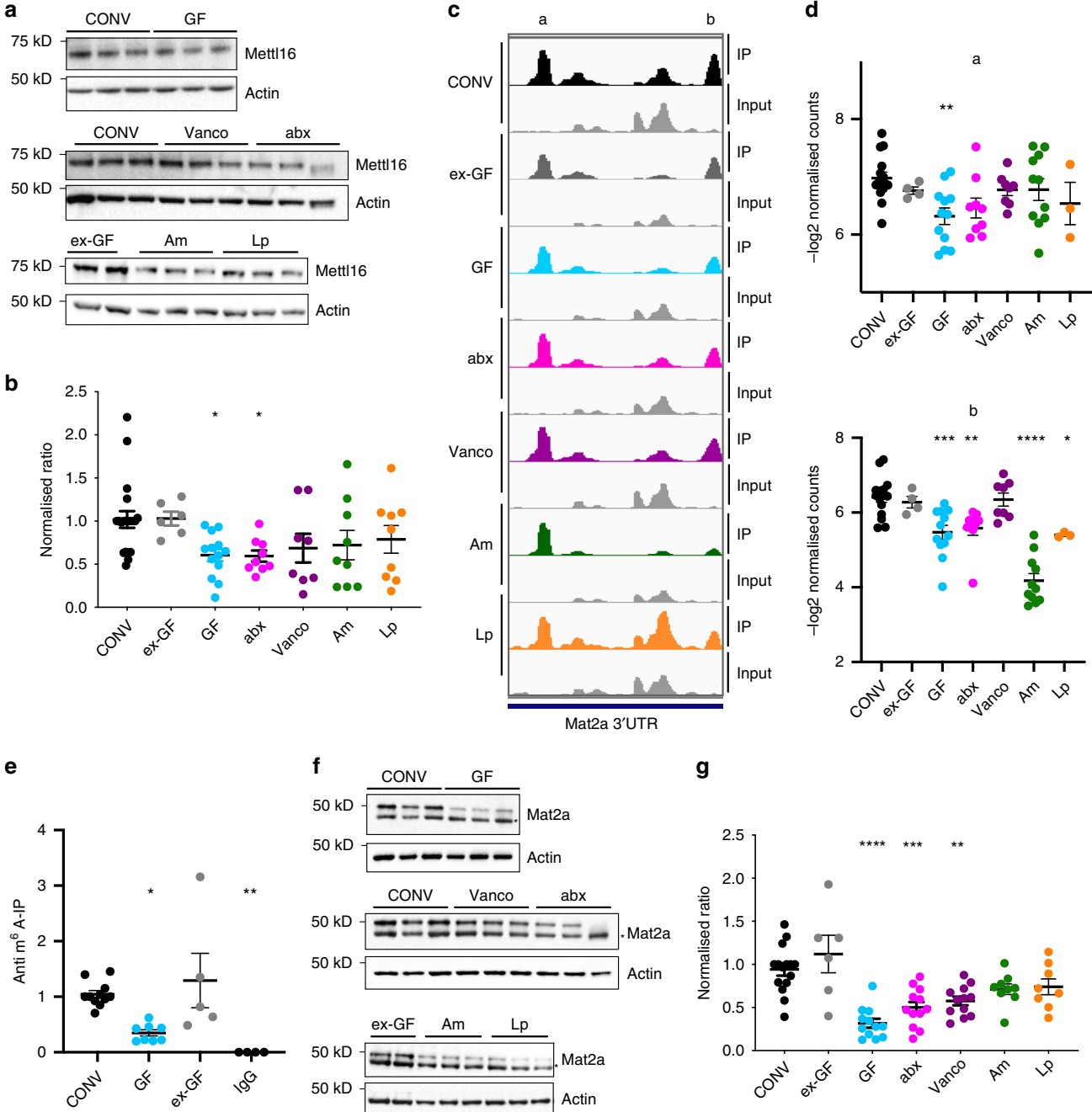

## Discussion

The microbiome has profound effects on host physiology, and regulates host gene expression on transcriptome, epigenome, and proteome level. Although more and more metabolites, such as short chain fatty acid (SCFA), amino acids, or bile acids are reported to mediate this influence, many aspects of the intricate interaction between commensal bacteria and their host remain elusive.

Here we show in a study of differential methylation of mRNA across a large number of biological conditions that the microbiota has a strong influence on epitranscriptomic m6A modifications in murine cecum. We found that changes in m6A modification profiles between conventional and germ-free mice can be restored by colonizing mice with a conventional specific pathogen free gut flora, and that the epitranscriptome changes could be induced by depleting the gut flora, or by enriching specific bacterial species

using antibiotics treatment. In their recent study, Wang et al.[35] similarly addressed the question if the host m6A epitranscriptome could be influenced by the microbiota by comparing methyltransferase expression and profiling of m6A modifications via MeRIP-Seq in a restricted number of conventional specific pathogen-free and germ-free mice[35]. They also found changes in the methylation profiles in intestine and liver, which were in a similar range concerning numbers of differentially methylated peaks as in the present study. However, Wang et al.[35] focused their study on the brain, since they detected strong changes in the expression of the methyltransferase Mettl3 in this organ, along with higher amounts of m6A nucleosides. In their study, Mettl3 expression was also slightly increased in the liver, and m6A nucleoside levels were higher in the intestine of GF mice. In our study we did not detect significant changes in Mettl3 protein levels in the brain or any other organ we analyzed (Supplementary

**Fig. 4 Differential expression of Mettl16 and Mat2a in mouse cecum. a** Western blot analysis of Mettl16 expression from ceca of CONV, GF, ex-GF, abx, vanco, Am, and Lp mice. Actin served as loading control. The membrane and thus the actin loading control for the blot displaying samples from ex-GF, Am, and Lp mice is identical to the western blot in Supplementary Fig. 4a. **b** Quantification using six different western blots; CONV $n = 20$, ex-GF $n = 6$; GF $n = 13$; abx $n = 9$; vanco $n = 8$; Am $n = 9$; Lp $n = 9$; Mettl16/Actin ratio was normalized to ratio in CONV mice in order to compare protein expression across multiple Western Blots. Ordinary one-way ANOVA was performed. *$p$-value < 0.05; $p$-values (compared to CONV): ex-GF: 0.9518; GF: 0.0255; abx: 0.0431; vanco: 0.1750; Am: 0.1783; Lp: 0.2728) (Holm–Sidak's multiple comparisons test). **c** Representation of two m6A peaks (mean of read per million normalized coverage (RPM) in detected methylation peaks) from anti-m6A immunoprecipitates and input in the 3'UTR of the Mat2a transcript in cecum. The peaks designated a and b were visualized for the indicated mouse models using IGV; **d** Quantification of indicated Mat2a peaks (a, b from b) as -log2 normalized read counts. Ordinary one-way ANOVA for multiple comparisons was performed. CONV ($n = 15$), GF ($n = 12$); ex-GF ($n = 4$); abx ($n = 9$); vanco ($n = 8$); Am ($n = 11$), Lp ($n = 3$); two independent sequencing experiments; a $p$-values (all compared to CONV): ex-GF: 0.6458; GF: 0.0039; abx: 0.0572; vanco: 0.6458; Am: 0.6458; Lp: 0.4695; b $p$-values (all compared to CONV): ex-GF: 0.889955; GF: 0.000323; abx: 0.003604; vanco: 0.889955; Am: 0.00000000000028; Lp: 0.018424; **e** anti-m6A-IP and qRT for Mat2a transcript in CONV ($n = 11$), ex-GF($n = 5$) and GF ($n = 8$) cecal RNA; IgG-IP ($n = 4$) served as control for unspecific binding; ordinary one-way ANOVA was performed; *$p$-value < 0.05; ***$p$-value < 0.005; $p$-values (compared to CONV): ex-GF: 0.6897; GF: 0.0127; IgG: 0.0029; **f** Western Blot analysis of Mat2a expression in CONV, GF, ex-GF, abx, vanco, Am, and Lp cecum. Actin served as loading control. Asterisk marks an unspecific band. **g** Quantification of Mat2a expression in CONV, GF, ex-GF, abx, vanco, Am, and Lp cecum. Quantification was performed using at least three different western blots for each condition; CONV $n = 15$, ex-GF $n = 6$; GF $n = 12$; abx $n = 12$; vanco $n = 12$; Am $n = 9$; Lp $n = 9$; Mat2a/Actin ratio was normalized to ratio in CONV mice in order to compare protein expression across multiple western blots; ordinary one-way ANOVA was performed. ***<0.0001, **$p$-value < 0.005 (Holm–Sidak's multiple comparisons test); $p$-values (all compared to CONV): ex-GF: 0.157; GF: 0.00000019; abx: 0.0002; vanco: 0.0019; Am: 0.1111; Lp: 0.154; data are presented as mean values ±SEM throughout the figure; details for statistical analysis and original data for **a**, **b**, **d**–**g** are given in the source data file.

Fig. 1d, Supplementary Fig. 4b, c), and its mRNA levels were only changed in the small intestine (Supplementary Fig. 1b). Consistent with the lack of Mettl3 expression changes, we did not detect higher amounts m6A by LC-HRMS (Supplementary Fig. 6) in cecum or liver. This is not surprising, considering the rather low number of differentially methylated peaks compared to the total number of peaks detected. It is unclear, if changes in mouse age, sex, or the composition of the conventional gut flora can explain the minor differences between the two studies. Nevertheless, the presence of a microbiota clearly affected the host m6A epitranscriptome in both studies to a similar extent, confirming the robustness of this finding.

Interestingly, we detected small changes in the protein expression of the methyltransferase Mettl16 in the cecum (Fig. 4a). By comparing our list of peaks differentially methylated between GF and CONV mice to known targets of METTL16 (Supplementary Fig. 5), we identified the S-adenosylmethionine synthase Mat2a to be among our differentially methylated transcripts. Mat2a was hypomethylated in GF mice, and, in a different affected peak, also in Am and Lp mice, whereas m6A peaks were similar to CONV in ex-GF, and vanco mouse cecum (Fig. 4b, c). Although the exact role of Mat2a m6A modification is under debate[5–7], the reduced protein levels of Mat2a that we detected in the cecum of GF and abx mice (Fig. 4e), might be linked to changed methylation patterns. S-adenosylmethionine (SAM) is the universal methyl donor used not only for RNA-methylation, but also for methylation of proteins, lipids, or DNA[59]. Changes of SAM levels that are influenced by nutrition, but also the microbiota, have been shown to influence DNA-methyltransferase activity, leading to changes in epigenetic marks[60]. Our study suggests that a similar regulation could be also the case for RNA-modifications.

We furthermore identified two commensal bacteria, *Akkermansia muciniphila* and *Lactobacillus plantarum* to influence host m6A modifications in the cecum and liver, both affecting cellular growth and proliferation, as well as cell death (Fig. 3e). Interestingly, *L.plantarum* has been previously shown to influence longevity and growth[54,55]. In the liver, metabolic pathways were most affected by *L.plantarum*. Considering the number of differentially methylated peaks in both liver and cecum, the influence of *L.plantarum* on host m6A modifications seems to be stronger than the influence of *A.muciniphila*. *L.plantarum* might influence SAM levels by providing the host with folate, which is

required for SAM synthesis[61,62]. However, it should be kept in mind, that the lower number of samples for Lp mice may lead to a higher number of detected differentially methylated peaks, as it is typically the case for transcriptomic studies[50]. Since the residual bacterial species persisting in the intestinal content in vanco mice seemed to be able to maintain a methylation profiles for many m6A peaks that was similar to CONV mice (Figs. 3d, e and Fig. 4b, c), we focused on two bacteria enriched in vanco mice for our monocolonisation experiments to test if these could restore methylation patterns similar to CONV mice. In addition, it would have been very interesting to monocolonize GF mice with bacteria, that were depleted in vanco mice to test whether these have less pronounced effects on methylation profiles than *A.muciniphila* and *L.plantarum*. Unfortunately, it is very difficult or even impossible to retrospectively include additional conditions in MeRIP-Seq experiments, as it is the case for other high throughput techniques such as RNA-sequencing or epigenetic analyses[63], which is a limitation of this study. Along this line, it would have been interesting to see if treatment of GF mice with antibiotics could explain the few differentially methylated peaks between GF and abx mice in the cecum, or if it is rather the few residual bacterial species surviving after antibiotics treatment that are responsible for this effect. We performed this experiment, but were not able to include this condition due to strong batch effects.

Differentially methylated peaks between GF and CONV mice in the cecum were mainly linked to metabolism and immunological and inflammatory responses, functions known to be influenced by the microbiota. Additionally, changes in m6A modifications of specific transcripts have been linked to inflammatory intestinal disease in humans[34], and also a T-cell specific conditional knock-out of Mettl3 led to intestinal inflammation in a mouse model[14]. Interestingly, we identified several risk genes for inflammatory bowel disease to be differentially methylated when comparing CONV and GF mice[64,65], suggesting that microbiota-induced changes in m6A modifications might thus be linked to inflammatory conditions.

Since the expression of m6A writer and eraser enzymes is very high in the spleen and presumably in immune cells in general (Supplementary Fig. 1a, b)[14], it will be very interesting to further study the role of microbiota-induced differential methylation of transcripts in these cell types. For the present study, however, we chose to investigate tissues in which, although affected by the

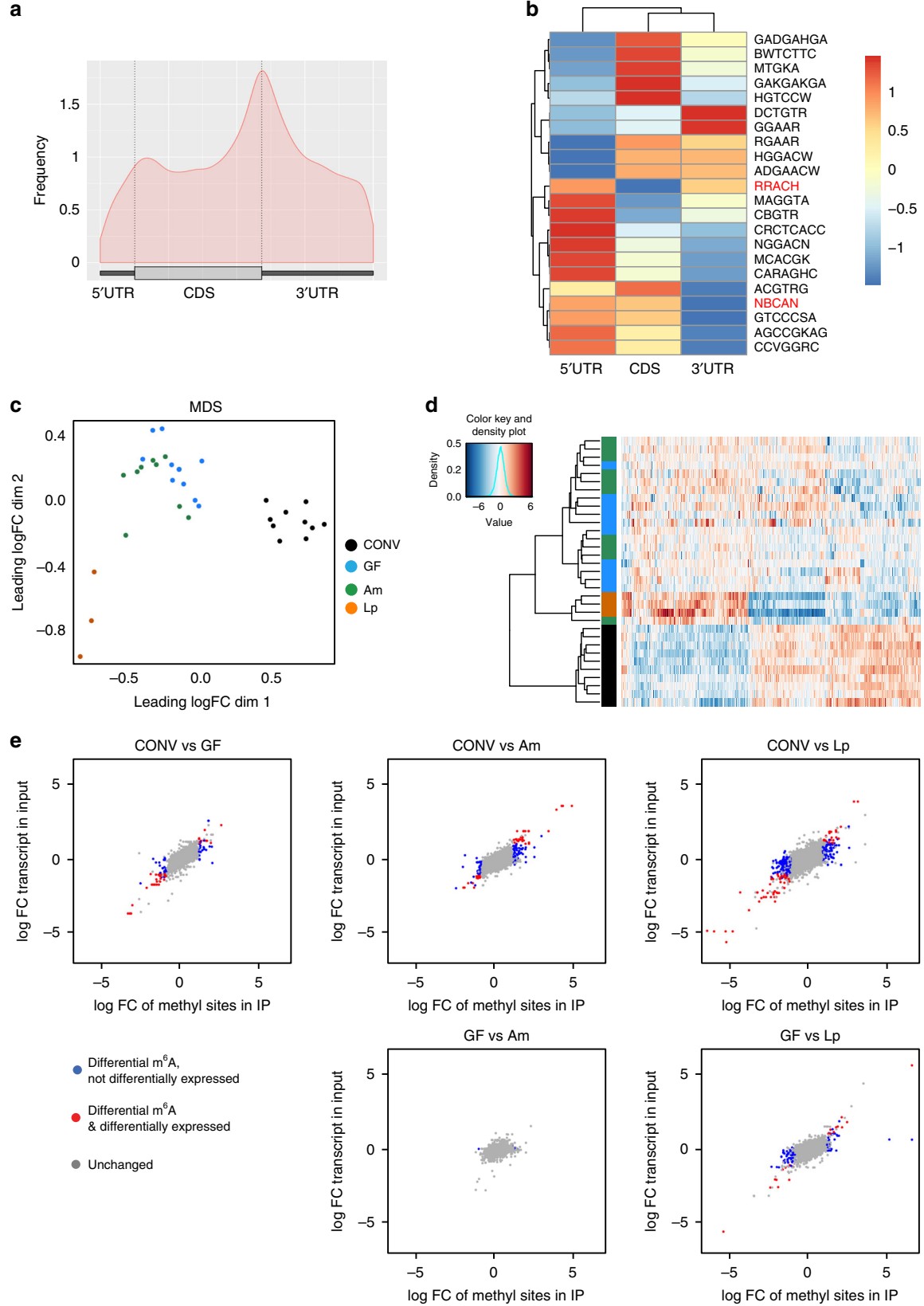

microbiota, numbers of cell types are not significantly changed in the absence of a microbiota, which is the case for many immune cells[37].

Taken together, epitranscriptomic modifications represent a novel mechanism of commensal-host-interaction in the intestine,

but also in the liver and other tissues[35], setting the ground for future studies on the regulation of a microbiota-directed methylation machinery and the translational consequences of this regulatory process as well as the effect of pathogens in the epitranscriptomic regulation.

**Fig. 5 m⁶A modifications in the liver. a** Positions of detected m6A peaks on all methylated transcripts in liver were determined and visualized using the GUITAR package. **b** Motif enrichment in m6A modification was determined by calculating total occurrence of motifs in m6A peaks on the 5'UTR, CDS and 3'UTR of in the liver. Consensus motifs for m6A (RRACH) and m6Am (NBCAN) are highlighted in red. **c** Multidimensional scaling (MDS) plot of the peak log counts-per-million IP data of all differentially methylated peaks showing the positions of the samples in the space spanned by the first and second MDS dimensions. Samples are colored with respect to condition. CONV ($n = 10$), conventionally raised mouse (black); GF ($n = 10$), germ-free mouse (cyan); Am ($n = 10$), *A.muciniphila*- monocolonized mice (green), Lp ($n = 3$), *L.plantarum*-monocolonized mice (orange). **d** Heat map of the peak log counts-per-million IP data based on all differentially methylated peaks, hierarchical clustering was performed using euclidean distance and ward.D2 linkage. **e** m6A-peaks found to be differentially methylated compared to differential expression of transcripts in indicated mouse models. Differentially methylated peaks that are also differentially expressed on transcript level are in red, differentially methylated peaks that are unchanged on transcript level, are in blue. Methylation peaks that were not significantly changed are shown in gray. *p*-values were estimated from moderated *t*-statistics with empirical Bayes moderation using limma package, followed by Benjamini-Hochberg correction. Cut-offs for differential expression are log fold change (FC) −1 to 1 and Benjamini–Hochberg- corrected *p*-values < 0.05.

## Methods

**Bacterial strains and culture conditions**. *Akkermansia muciniphila* (ATCC BA-835), obtained from the Biological Resource Center of Institut Pasteur (CRBIP), was cultured in Brain Heart Infusion (BHI) supplemented with 8mM L-Cysteine hydrochloride, 0.2% NaHCO₃, and 0.025% Hemin in an anaerobic atmosphere using Oxoid™ AnaeroGen™ 2.5L gas packs (Thermo Fisher) at 37 °C. *Lactobacillus plantarum* (BAA-793) was obtained from ATCC and cultured in MRS broth (Thermo Fisher) at 37 °C and 5% CO₂.

**Animal experiments**. All animal experiments were approved by the committee on animal experimentation of the Institut Pasteur and by the French Ministry of Agriculture.

**Mice**. C57BL/6J mice were purchased from Charles River. Germ-free mice generated from C57BL/6J mice were obtained either from CNRS TAAM UPS44 Orléans, France, or from the Gnotobiology Platform of the Institut Pasteur and kept in isolators. Conventional mice were kept in specific pathogen- free conditions and all the mice used were female. Mice were housed in 10 h (dark)/ 14 h (light) cycles.

**Colonization of germ-free mice**. For generating ex-GF mice, 10–12 fecal pellets were collected from 4 CONV mice housed in the same cage for 2 weeks, resuspended, and added to the drinking water of GF mice. This procedure was repeated on 3 consecutive days. On Day 4, cecal content of the 4 CONV mice was collected, resuspended in 10 mL of PBS, of which 0.2 mL was administered to the pre-colonized mice by oral gavage. Colonization efficiency was determined in the cecal content of mice after 4 weeks. For mono-colonization experiments, germ-free mice were inoculated once with 10⁹ colony-forming units (CFU) of *A. muciniphila* or *L.plantarum* and maintained for four weeks. Mono-colonization was monitored by *A.muciniphila*-[66] or *L.plantarum*-[67] specific PCR using the following primers:
  AM1 (for) 5′ CAGCACGTGAAGGTGGGGAC 3′
  AM2 (rev) 5′ CCTTGCGGTTGGCTTCAGAT 3′
  Lp (for) 5′ AAT TGA GGC AGC TGG CCA 3′
  Lp (rev) 5′ GAT TAC GGG AGT CCA AGC GG 3′
and controlled by 16S rRNA sequencing in cecal content.

**Antibiotic treatment of mice**. For depletion of the gut microbiota, conventional C57BL/6J mice were treated with antibiotics as described previously[68]. In brief, after oral treatment with amphotericin B (0.1 mg/mL; Sigma Aldrich) for 2–3 days, mice were treated with a solution consisting of 10 mg/mL ampicillin, 5 mg/mL vancomycin 10 mg/mL neomycin, 10 mg/mL metronidazol, and 0.1 mg/ml amphotericin-B (all Sigma Aldrich) per os every 12 h using a gavage volume of ~10 mL/kg body weight for 21 days. The depletion was controlled for by quantitative PCR as detailed below and the identity of residual bacterial genera determined by 16S rRNA sequencing. For enrichment of a small number of genera, mice were treated as above with only 5 mg/mL vancomycin and 0.1 mg/ml amphotericin-B. Tissues used for the analysis were derived from 3–4 independent experiments.

**RNA preparation, ribodepletion, and mRNA purification**. Total RNAs were prepared using the RNeasy maxi kit (Qiagen) and quality-controlled using RNA 6000 Nano assay (Agilent). Ribodepletion was performed using RiboMinus Transcriptome Isolation Kit (Human/Mouse; Thermo Fisher). mRNA purification was performed using Dynabeads® mRNA Purification Kit for mRNA Purification from Total RNA preps (Thermo Fisher). Absence of ribosomal RNA contaminations was controlled using RNA 6000 Pico assay (Agilent).

**LC-HRMS**. Ribodepleted RNAs were desalted using Microcon YM10 columns (Millipore) and subjected to Nuclease P1 digestion in 50 mM ammonium acetate (pH 7) in the presence of Antarctic phosphatase (New England Biolabs) as

described previously[69]. Nucleoside composition was analyzed by narrow bore HPLC using a U-3000 HPLC system (Thermo-Fisher). An Accucore RP-MS (2.1 mm × 100 mm, 2.6 µm particle) column (Thermo-Fisher) was used at a flow rate of 200 µl/min at a temperature of 30 °C. Mobile phases used were 5 mM ammonium acetate, pH 5.3 (Buffer A) and 40% aqueous acetonitrile (Buffer B). A multilinear gradient was used with only minor modifications (5 min plateau at 100% instead of 3 min) from that described previously[70].

An LTQ Orbitrap™ mass spectrometer (Thermo Fisher Scientific) equipped with an electrospray ion source was used for the LC/MS identification and quantification of nucleosides. Mass spectra were recorded in the positive ion mode over an *m/z* range of 100–1000 with a capillary temperature of 300 °C, spray voltage of 4.5 kV and sheath gas, auxiliary gas and sweep gas of 40, 12, and 7 arbitrary units, respectively. Calibration curves were generated using a mixture of synthetic standards of Adenosine (A) and Cytidine (C) (Sigma-Aldrich), m6A, and m1A, m5C (TCI Europe), and N6, 2′-O-dimethyladenosine (m6Am) (Berry & Associates) in the ranges of 20–625 injected fmol for m1A, m6A, m6Am, and m5C and 5–250 injected pmol for A and C. Each calibration point was injected in triplicate. Extracted Ion Chromatograms (EIC) of base peaks of the following masses: A (*m/z* 268.08-268.12), C (*m/z* 244.08-244.11), m1A and m6A (*m/z* 282.10-282.13), m6Am (*m/z* 296.12-296.15), and m5C (*m/z* 258.09-258.12), were used for quantification. In all cases, coefficients of variations for peak areas were always below 15%. Experimental data (peak area versus injected quantity) were fitted with a linear regression model for each compound leading to coefficient of determination ($R^2$) values better than 0.997. Accuracies were calculated for each calibration point and were always better than 15%.

**Western blotting**. For Western blotting, we performed acetone precipitation of the flow-through of the tissue lysate from RNA preparation according to the manufacturer's instructions. Samples were solubilized, separated on NuPAGE (Thermo Fisher) or criterion (Bio-Rad) 4–12% Bis-Tris Protein Gels and transferred to PVDF membranes using iBlot (Thermo Fisher). Antibody dilutions were: rabbit anti Mettl3 (abcam ab195352), rabbit anti Mettl16 (abcam ab186012), rabbit anti Alkbh5 (abcam ab195377) and rabbit anti Mat2a (abcam ab154343): 1:1000; mouse anti β-actin (Sigma-Aldrich A1978): 1:2000; rabbit anti Mettl14 (Sigma Aldrich HPA038002): 1:500.

**Proteomics analysis**. Protein digestion: precipitated proteins were solubilized with 8 M urea in Tris HCl 50 mM pH 8.5. Proteins disulfide bonds were reduced with 5 mM tris (2-carboxyethyl)phosphine (TCEP) for 20 min at 23 °C and further alkylated with 20 mM iodoacetamide for 30 min at room temperature in the dark. Subsequently, LysC (Promega) was added for the first digestion step (protein to Lys-C ratio = 80:1) for 3 h at 30 °C. Then the sample was diluted to 1 M urea with 100 mM Tris pH 8.0, and trypsin (Promega) was added to the sample at a ratio of 50:1(w/w) of protein to enzyme for 8 h at 37 °C. Proteolysis was stopped by adding 1% formic acid (FA). Resulting peptides were desalted using Sep-Pak SPE cartridge (Waters) according to manufacturer's instructions. Peptides elution was done using a 50% acetonitrile (ACN), 0.1% FA buffer. Eluted peptides were stored until use.

Mass spectrometry analysis: peptides were analyzed on a Q Exactive Plus instrument (Thermo Scientific) coupled with an EASY nLC 1000 chromatography system (Thermo Scientific). Samples were loaded on an in-house packed 50 cm nano-HPLC column (75 µm inner diameter) with C18 resin (1.5 µm particles, 100 Å pore size, Reprosil-Pur Basic C18-HD resin, Dr. Maisch GmbH) and equilibrated in 98% solvent A (H₂O, 0.1% formic acid) and 2% solvent B (acetonitrile, 0.1% formic acid). Peptides were eluted using a gradient of solvent B (ACN, 0.1% FA) from 3 to 6% in 5 min, 6 to 29% in 130 min, 29 to 56% in 26 min, and 56 to 90% in 5 min (total length of the chromatographic run was 180 min including high ACN level steps and column regeneration). The instrument method for the Q Exactive Plus was set up in the data dependent acquisition mode. MS spectra were acquired at a resolution of 70,000 (at *m/z* 400) with a target value of $3 \times 10^6$ ions. The scan range was limited from 300 to 1700 *m/z*. Peptide fragmentation was performed using higher-energy collision dissociation (HCD) with the energy set at 27 NCE. Intensity threshold for ions selection was set at

$1 \times 10^6$ ions with charge exclusion of $z = 1$ and $z > 7$. The MS/MS spectra were acquired at a resolution of 17,500 (at $m/z$ 400). Isolation window was set at 1.6 Th. Dynamic exclusion was employed within 45 s.

Data processing and analysis: all data were searched using Andromeda[71] with the MaxQuant software version 1.5.3.8[72,73] against Uniprot proteome database of mouse. Usual known mass spectrometry contaminants and reversed sequences were also searched. Andromeda searches were performed choosing trypsin as specific enzyme with a maximum number of two missed cleavages. Possible modifications included carbamidomethylation (Cys, fixed), oxidation (Met, variable), N-terminal acetylation (variable), and ubiquitin (variable). The mass tolerance in MS was set to 20 parts per million (ppm) for the first search then 6 ppm for the main search and 10 ppm for the MS/MS. Maximum peptide charge was set to 7 and 5 amino acids were required as minimum peptide length. The "match between runs" feature was used between condition with a maximal retention time window of 1 min. One unique peptide to the protein group was required for the protein identification. A false discovery rate (FDR) cutoff of 1% was applied at the peptide and protein levels. The MaxLFQ, Maxquant's label-free quantification (LFQ) algorithm was used to calculate protein intensity profiles across samples[71]. Data were filtered by requiring a minimum peptide ratio count of 2 in MaxLFQ. The mass spectrometry proteomics data have been deposited to the ProteomeXchange Consortium via the PRIDE partner repository with the dataset identifier PXD016099.

Statistical analysis: for the statistical analysis of one condition versus another, proteins identified in the reverse and contaminant databases and proteins only identified by site were first discarded from the list. Then, proteins exhibiting fewer than 2 LFQ intensities in at least one condition were discarded from the list to avoid misidentified proteins. After log2 transformation of the residual proteins, summed intensities were normalized by median centering within conditions (normalizeD function of the R package DAPAR[74]). Remaining proteins without any LFQ intensities in one of the two conditions have been considered as proteins present in one condition and absent in another. They have therefore been set aside and considered as differentially abundant proteins. Next, missing values were imputed using the impute.MLE function of the R package imp4p. Proteins with a log2 (fold-change) inferior to 1 have been considered as proteins which are not significantly differentially abundant. Statistical testing of the remaining proteins (having a log2 (fold-change) superior to 1) was performed using the limma $t$-test within the R package limma[75]. An adaptive Benjamini–Hochberg procedure was applied on the resulting $p$-values using the function adjust.p of R package cp4p to estimate the proportion of true null hypotheses among the set of statistical tests[76]. The proteins associated to an adjusted $p$-value inferior to an FDR level of 1% have been considered as significantly differentially abundant proteins. Finally, the proteins of interest are therefore those emerging from this statistical analysis in addition to those which are considered to be absent from one condition and present in another.

**Immunoprecipitation of m6A-methylated mRNA**. Immunoprecipitation was performed as previously described[77]. Briefly, 3 µg of rabbit anti m6A antibody (Synaptic Systems) were bound to 25 µl washed Protein G Dynabeads (Thermo Fisher) in immunoprecipitation buffer (1x IPP; 150 mM NaCl, 0.1% NP-40, 10 mM Tris-Cl, pH 7.4) for 30 min at room temperature (RT) and washed twice in 1x IPP. Total (liver), ribodepleted (cecum) or polyA-selected (cecum & liver) RNA was fragmented using the NEBNext Magnesium RNA fragmentation module (New England Biolabs), purified by ethanol precipitation and quality-controlled using the RNA 6000 pico assay (Agilent). Equal amounts of RNA (5 µg for ribodepleted and polyA-selected RNA and 200 µg for total RNA) were denatured for 2 min at 70 °C and adjusted to 1x IPP concentration using 2x IPP. RNA was added to the antibody-bound beads and incubated for 2 h at 4 °C in the presence of murine RNase inhibitor (New England Biolabs). The bound RNA was washed twice with 1x IPP, twice with low salt IPP buffer (50 mM NaCl, 0.1% NP40, 10 mM Tris-Cl pH 7.4), twice with high salt IPP buffer (500 mM NaCl, 0.1% NP40, 10 mM Tris-Cl pH 7.4) and once more in 1x IPP. RNA was eluted using 30 µl of buffer RLT (Qiagen). In all, 20 µl of MyOne Silane Dynabeads (Thermo Fisher) were washed with RLT and resuspended in 30 µl of RLT. Eluted RNA was bound to the beads in the presence of 35 µl of absolute ethanol, washed twice in 70% ethanol and eluted in 100 µl of $H_2O$. RNA was purified and concentrated using RNA clean & concentrator (Zymo research) before proceeding to library preparation. If subsequent qRT PCR analysis was performed, RNA fragmentation was omitted and the cDNA synthesis performed using the Quantitect reverse transcription kit (Qiagen). For Mat2a qRT PCR, TaqMan® Universal Master Mix II, with UNG (Thermo Fisher) was used. The qPCR probe for Mat2a (Mm.PT.58.8961804) was obtained from IDT. For GFP qRT-PCR, Evagreen SsoFast and the following primers were used:
GFP (for): 5′ ATGGTGAGCAAGGGCGAGGAG 3′;
GFP (rev): 5′ TTGTACAGCTCGTCCATGCCG 3′.

**qRT PCR**. For qRT PCR, 1 µg of RNA was retrotranscribed using the Quantitect cDNA reverse transcription kit (Qiagen). Quantitative real-time PCR was performed using PrimeTime® Gene Expression Master Mix and the following probes (all IDT): Mettl3 (Mm.PT.58.10309074), Mettl14 (Mm.PT.58.33500780), Mettl16 (Mm.PT.58.9941116), Alkbh5 (Mm.PT.58.6928987), Fto (Mm.PT.58.32888407),

and Pcif1 (Mm.PT.58.9572065). TaqMan probes for housekeeping genes Gapdh (Mm99999915_g1) and Hprt (Mm03024075_m1), were from Thermo Fisher.

**In vitro transcription of GFP RNA**. GFP coding sequence was cloned into pCRII and the plasmid linearized using BamHI. In vitro transcription was performed using the Maxiscript T7 kit (Thermo Fisher) according to the manufacturer's instructions, but replacing ATP with N6-Methyl-ATP (Trilink). The template was digested using Turbo DNase and RNA purified using RNA clean & concentrator-5 (Zymo).

**Library preparation and sequencing**. In all, 150 ng of input RNA and the immunoprecipitated RNAs were dephosphorylated using Antarctic phosphatase and subjected to T4 PNK treatment (both New England Biolabs). Directional RNA-seq libraries were prepared using NEBNext Multiplex Small RNA Library Prep Set for Illumina (New England Biolabs) according to the manufacturer's instructions. Libraries were sequenced on an Illumina HiSeq 2500 platform generating single end reads (65 bp).

**MeRIP-Seq processing**. The mouse mm10 genome and list of transcripts were downloaded from Gencode (*Mus Musculus* VM13[78]). Only the 21968 'protein_coding' genes were kept for the analysis. After the sequencing of every MeRIP-Seq (IP dataset) and RNASeq (Input dataset) the resulting reads were trimmed (AlienTrimmer 0.4.0[79], default parameters). They were mapped on mouse mm10 genome using STAR mapper 2.5.0a (–sjdbOverhang 100 parameter)[80]. Mapping files were filtered to keep uniquely mapped reads using SAMtools 0.1.19 (samtools view -b –q 1 parameters)[81], and saved to BAM files after indexation. The quality of the sequencing and mapping was assessed using FastQC 0.10.1 and MultiQC 0.7[82,83]. Gene expression was calculated with HTSeq 0.9.1(-s no -m union --nonunique all parameters)[84].

**m6A peak detection**. The original three reference papers on MeRIP-Seq analysis have used three different workflows for m6A modification peak detection (Fisher test[2], MACS2 software[85], and Peak Over Input technique[77]). Each of these techniques has a certain bias and identifies different types of methylation peaks. We implemented all of them https://gitlab.pasteur.fr/hub/MeRIPSeq and developed our own technique using the fold of Reads Per Million (RPMF). We prepared the peak detection by first generating windows of 100 bp overlapping in their middle using all 21,968 'protein_coding' genes. Windows of 100 bp overlapping in their middle were generated. The total number of reads per window was calculated for each dataset. The number of reads for each window in IP and Input was determined using HTSeq 0.9.1[84](-s no -m union --nonunique all parameters). Only the windows with coverage higher than 10 reads in IP datasets were kept. Fisher, POI, and RPMF techniques were then run on these windows to assess for the presence of methylation peaks.

The Fisher exact rank test was applied on each of the 100-bp windows. The $p$-values were corrected with Benjamini-Hochberg multiple testing correction[86]. Only peaks with a corrected $p$-value < 0.001 were kept for Fisher analysis. The reads per million (RPM) for each window in IP and Input was calculated. The reads per million-fold (RPMF) was determined by subtraction of RPM of each window in the IP dataset and RPM in the Input. Only the peaks with RPMF > 10 were kept. We prepared the POI peak detection by first calculating the raw coverage on each dataset, IP and Input, using BEDTools 2.17.0[87] (genomcov -d -split -ibam) and removing all positions with null coverage. Following the workflow described in Schwartz et al.[77], the Peak over median (POM) was calculated by dividing median expression in the window by median expression of the gene, only considering exonic regions of genes. The peaks with a POM score <4 in the IP dataset were removed. The Peak Over Input (POI) score was then calculated by dividing POM score in the IP dataset by POM score in the Input dataset. Only the peak with POI >2 were kept. MACS2 methylation sites detection was run (-g 282000000 –nomodel parameters) on bam files of IP datasets with the Input datasets serving for assessing the whole RNA distribution. Each of the four methods of peak detection was detecting small windows with potential methylation sites in very few datasets. To keep only robust methylation sites, the occurrence of each peak was assessed by counting in how many IP samples a specific window was detected as a methylation site. Only windows found in three or more IP samples were kept. BEDtools merge software[87] was run to regroup all overlapping windows. A Median coverage across all IP samples was calculated using WiggleTools[88]. With an in-house python script, the maximum coverage position of every peak for every technique was computed and the peak region was redefined as the 150 bp region centered at the position of maximum coverage. Methylation sites found in at least three of the four techniques were combined using BEDTools merge function, followed by another search for maximum coverage for each site. Finally, a general list of 150-bp-long peaks centered at the maximum coverage in every IP dataset was extracted. The overlapping region on their corresponding gene was searched: 5′UTR, 3′UTR, CDS, intron. The presence of the m6A consensus sequence RGACW in the sequence of the peak was assessed by calculating a motif score by adding score presence when one of this sequence was found: 'GAACA': 2, 'GGACA': 3, 'GAACT': 5,'GGACT': 8. The union of all the methylation sites found by each of the four techniques was determined using

BEDTools merge function. Two lists of potential methylation sites were then extracted: 37,796 methylation peaks for the cecum and 31,394 peaks for the liver tissue. We determined the overlap of these peaks with the m6A modification peaks previously detected by MeRIP-Seq stored in the MeT-DB v2.0 database[39] using the Bioconductor R package GenomicFeatures[89].

**Differential methylation and expression analysis**. Statistical analyses were performed using the R package limma[75]. https://gitlab.pasteur.fr/hub/MeRIPSeq. Read count data were first normalized with the trimmed mean of M-values normalization method[90] (edgeR package) and transformed with the voom[91] function (limma package). Limma was then used to assess the statistical significance of observed differences in read counts. Two different linear models were derived to address two different questions. Differentially expressed genes between all pairs of the four conditions were first detected with $y \sim BioCond + Sequencing + Library\_batch + IP\_procedure$, where y is the normalized and transformed Input read counts (expression data), BioCond refers to the seven (CONV, GF, ex-GF, abx, vanco, Am, Lp) biological conditions under study for cecum, or 4 (CONV, GF, Am, Lp) biological conditions under study for liver. Batches associated with sequencing, library preparation, and IP procedure were also accounted for through the $Sequencing$, $Library\_batch$, and $IP\_procedure$ variables. Differentially expressed methylation peaks were derived using a model on IP read counts to detect differential methylation ($y \sim BioCond + Sequencing + Library\_batch + IP\_procedure$). Only genes and peaks with at least 5 counts per million (CPM) in at least three samples were included in the differential analyses. In addition, to avoid biases caused by the two different IP procedures, only peaks with at least 5 CPM in at least two IP and input samples of each sequencing procedure per condition were included in the differential analysis (19,061 peaks for the cecum and 10,383 peaks for the liver). p-values resulting from the two models were adjusted for multiple comparisons using the BH procedure[86]. Genes or methylation peaks were considered statistically different when their adjusted p-value was lower than 0.05 and the absolute log-fold change was >1. The batch effects were removed from the data for plotting the multidimensional scaling (MDS) and heat maps plots using the removeBatch-Effect function of the limma package. The MDS plots were done with the function plotMDS. The GUITAR plot was generated using the GUITAR package[39]. Pathway analysis was performed using the Ingenuity Pathway analysis software (IPA; Qiagen) using the Diseases & Functions tool after performing expression analysis of differentially methylated peaks with the same parameters as above (absolute log-fold change >1, p-value < 0.5). For pathways analysis of differentially methylated transcripts in liver, KEGG mouse 2019 and GO Biological Process analyses in Enrichr were used[92,93].

**Differential transcript usage analysis**. Using the input samples, we performed a differential transcript usage (DTU) analysis by following the workflow described in Love et al.[94]. Isoform abundance estimates were computed with salmon v0.13.1[95] and scaled transcript-per-million estimates were imported with tximport v1.10.1[96]. We performed a differential isoform usage analysis, using the same covariables as in the differential expression analysis, computed a per-gene adjusted p-value using DEXSeq v1.28.3[97] and used stageR v1.5.1[98] to compute overall false discovery rate per transcript. Isoforms were considered to be differentially used between two conditions if their overall FDR was smaller than 5% and their change in isoform fraction between the two conditions was larger than 0.1.

**Motif presence**. MEME-ChiP software[99] was used to search for motif in different lists of m6A peaks. For each tissue we used: List of all m6A peaks (e.g. Liver), list of non-differentially methylated peaks (e.g. Liver_NoMeth), list of differentially methylated peaks (e.g. Liver_Meth), list of differentially methylated peaks with corresponding gene differentially expressed (e.g. Liver_MethGene), list of differentially methylated peaks overlapping a gene which is not differentially expressed (e.g. Liver_MethNoGene). MEME-ChiP runs different motif search software from the MEME-Suite, such as DREME[100] and CentriMo[101], and centralizes the results in one folder. Using in-house python scripts, all motifs fund by DREME were combined. Only motifs with DREME e-value <0.00001 and found centrally enriched by CentriMo were kept. Previously described motifs were added in the motif list: RRACH, NGGACN, and NBCAN[1,2,26]. To avoid motif presence list size effect induced by small list of peaks, only lists of peaks with >50 m6A peaks were kept in the motif presence analysis. For motif presence we added the list of all peaks sorted by their overlapping region on a transcript (e.g. Liver_CDS, Liver_3UTR, Liver_5UTR). The cumulative occurrence of each motif was calculated for each list and divided by the number of m6A peaks per list. Each row of the final table obtained was then reduced and centered to perform clustering of the motif presence with in-house R scripts.

**qPCR from cecal content**. gDNA was prepared from cecal content using the Power Soil DNA isolation kit (MoBio) following the manufacturer's instructions.

qPCR was performed as described[68] using EvaGreen Sso Fast Master Mix (Biorad), 500 nM primers and 50 ng input gDNA. Primer sequences were:

16S V2 (for) 5′AGYGGCGIACGGGTGAGTAA 3′
16S-V2 (rev) 5′CYIACTGCTGCCTCCCGTAG 3′
mpIgRgenomic (for) 5′ TTTGCTCCTGGG CCTCCAAGTT 3′
mpIgRgenomic (rev) 5′AGCCCGTGACTGCCACAAATCA3′.

Relative expression was determined by amplification of the 16S rRNA V2 region using the ΔΔCT method with mpI genomic region as a reference[68].

**16S rRNA sequencing**. 16S rRNA sequencing was performed as described previously[102]. Briefly, the 16S rRNA gene amplification was performed by using the Nextflex 16s v1-v3 amplicon-seq kit. gDNA isolated from cecal content DNA was sequenced by using Illumina Miseq. Library adapters, primer sequences, and base pairs occurring at 5′ and 3′ends with a Phred quality score <20 were trimmed off by using Alientrimmer (v0.4.0). Reads with a positive match against mouse genome (mm10) were removed. Filtered high-quality reads were merged into amplicons with Flash (v1.2.11). Resulting amplicons were clustered into operational taxonomic units (OTU) with VSEARCH (v2.3.4)[103]. The process includes several steps for de-replication, singletons removal, and chimera detection. The clustering was performed at 97% sequence identity threshold, producing 1110 OTUs. The OTU taxonomic annotation was performed with the SILVA SSU (v132) database[104] using VSEARCH[103] and filtered according to their identity with the reference[105]. Annotations were kept when the identity between the OTU sequence and reference sequence was ≥78.5% for taxonomic Classes, ≥82% for Orders, ≥86.5% for Families, ≥94.5% for Genera and ≥98% for species. In total, 56% of the OTUs were assigned a genus level annotation following these criteria. The input amplicons were then aligned against the OTU to get a contingency matrix giving the number of amplicons associated with each OTU using VSEARCH global alignment[103]. On average 91.5% of amplicons were aligned against the OTUs.

The contingency matrix normalization was performed at OTU level using the weighted non-null normalization, described in detail by Volant et al.[106]. Normalized counts were then summed within the genera. The generalized linear model (GLM) implemented in the DESeq2 R package[107] was then applied to detect differences in abundance of genera between each group. We defined a GLM that included the treatment (condition) and the time (variable) as main effects and an interaction between the treatment and the time. Resulting P values were adjusted according to the Benjamini and Hochberg procedure[86]. The results of the OTU picking process is available on shaman.pasteur.fr[106] with the key ff9551570bf15. The statistical analysis can be reproduced on SHAMAN by loading the datasets available on figshare: https://doi.org/10.6084/m9.figshare.8321165.v5. The bioinformatic workflow implemented in SHAMAN is available at github.com/aghozlane/masque. Normalized OTU tables are provided as Supplementary Data 6.

**Quantification and statistical analysis**. All data are expressed as mean and standard error of the mean. The number of animals or replicates (n) for each group is indicated in the figure legends. Either student's t-test or Ordinary one-way or two-way ANOVA for multiple comparisons were used for statistical analysis. This information is provided in the figure legends. For differential expression analysis p-values were adjusted using the Benjamini–Hochberg procedure as indicated in the figure legends. p-values < 0.05 were considered significant.

**Reporting summary**. Further information on research design is available in the Nature Research Reporting Summary linked to this article.

## Data availability

MeRIP-Seq and RNA-seq data have been deposited in the ArrayExpress database at EMBL-EBI under accession number E-MTAB-6560. 16S rRNA sequencing data have been deposited in the European Nucleotide Archive database at EMBL-EBI under accession number PRJEB25147. 16S data are also available in SHAMAN[106] (shaman. pasteur.fr; key ff9551570bf15) and Figshare (https://doi.org/10.6084/m9. figshare.8321165.v5). The mass spectrometry proteomics data have been deposited to the ProteomeXchange Consortium via the PRIDE[108] partner repository with the dataset identifier PXD016099. Proteomic data were searched against a uniprot database containing *Mus musculus* proteins (downloaded 03/2016; https://www.uniprot.org/ proteomes/); public databases used were: MeT-DB v2.0 database (http://180.208.58.19/ metdb_v2/html/index.php), SILVA SSU (v128) (https://www.arb-silva.de/ documentation/release-128/), Gencode (mouse mm10 genome and list of transcripts; Mus Musculus VM13; https://www.gencodegenes.org/mouse_releases/).

Source data for Fig. 4a, b, d–g, and Supplementary Figs. 1a–d, 2a, 3e, 4a–c, and 6 are given in the Source data file associated to this manuscript.

## Code availability

All the scripts used for m6A peak detection and methylation peak analysis have been deposited on Institut Pasteur GitLab (https://gitlab.pasteur.fr/hub/MeRIPSeq). The bioinformatic workflow for 16S sequencing analysis is available at github (https://github.com/aghozlane/masque).

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

## Acknowledgements

We thank Marion Bérard and the Gnotobiology platform of the Institut Pasteur for support in conducting gnotobiology experiments; the Pasteur Biomics platforms for Transcriptomics and Epigenetics and Metagenomics for providing RNA sequencing and 16S rRNA sequencing services; Cédric Fund, Biomics Platform, C2RT, Institut Pasteur, Paris, France, supported by France Génomique (ANR-10-INBS-09-09) and IBISA; Carla Saleh and Olivier Dussurget for helpful discussions; Hugo Varet (Institut Pasteur Bioinformatics and Statistics Hub) for advice concerning statistical analysis; Stevenn Volant for help with analysis of 16S sequencing data. This work was supported by grants to P.C.: European Research Council (ERC) Advanced Grant BacCellEpi (670823), ANR Investissement d'Avenir Programme (10-LABX-62-IBEID), ERANET-Infect-ERA PROANTILIS (ANR-13-IFEC-0004-02) and the Fondation Le Roch-Les Mousquetaires. S.J. was supported by a Danone Research fellowship and European Research Council (ERC) Advanced Grant BacCellEpi (670823).

## Author contributions

S.J. designed the study and conducted most of the experiments, analyzed data and wrote the manuscript; A.B. and C.B. performed bioinformatical and statistical analysis; M.A.N. helped with mouse experiments; A.G. analyzed 16S rRNA sequencing data; A.P. and G.S. helped with western blot analyses; V.G. and D.T. performed LC-HRMS experiments; T.C., and M.M. performed proteomics study; Q.G.G. performed statistical analysis of proteomics data; M.A.D. supervised bioinformatics and performed statistical analyses; P.C. designed the overall study and wrote the manuscript.

## Competing interests

The authors declare no competing interests.
