## [Peer Review File · Nature Communications]

Reviewers' comments:

Reviewer #1 (Remarks to the Author):

In this study, Jabs et. al present evidence for association between microbiota and host tissue's m6A RNA methylation. Extensive data resulting from multiple MeRIP experiments in cecum and liver across different mice models provides strong support for several claims in the study. However, there are several major concerns as summarized below making various aspects of study weak.

Major concerns

- 1) Manuscript title is very generic and reads like a review. Since the authors only focused on two tissues indicating different host tissues is not appropriate and justified. At the least the authors should extend the work to additional tissues or change the title to something more specific such as 'Impact of gut microbiota in on the m6A epitranscriptome in mouse cecum and liver tissues.'
- 2) In fact, it is unclear why the authors only limited to cecum and liver, the authors should at least provide the expression levels of methyltransferases across multiple mice tissues including brain to demonstrate a rationale for choosing these tissue types. For instance, it is increasingly known that immune tissues exhibit RNA methylation so including them to justify the choice would be appropriate.
- 3) The notion that MeRIP-seq also identifies m6Am is not well established and is still debatable. In addition, the authors neither show data or prior publication in support of this claim although this may indeed be the case in several loci for MeRIP-seq. So the authors should either cite a paper showing this evidence especially since m6Am positions have not been mapped accurately in mice tissues or provide data in support of this claim? In the absence of this, the authors should change throughout the manuscript m6A/m to m6A.
- 4) To show the reproducibility of the results across species and experimental systems, at the least the authors should demonstrate in human cell lines such as HepG2 (liver) in the presence of various commensal bacteria that m6A peaks are detected and provide a comparison of the overlap. Likewise, providing protein expression levels of Mettl's in human cell lines (with and without exposure to microbiota) should complement the claims.
- 5) The authors present data in Fig 2E that the METTL16 enzyme (methyltransferase) expression levels are different between CONV, mono-, germ-free mice and those in close contact with gut microbiota. However, no statistical data is shown, so it is unclear whether the trends are significant. Authors should present this data for all the METTLs analyzed for protein levels in both cecum and liver across the mouse models. In fact, to this reviewer it makes more sense to see this plot as one of the figure panels in Figure 1 before delving into MeTIP-seq data/results. In the absence of such differences in levels at RNA or protein, it is hard for this reviewer to know whether the differences are due to differential detection across samples or due to microbiota.
- 6) Would the data in Fig 1G still be the same if proteomic profiling is performed in couple of samples i.e, does differential methylation lead to differential protein levels in the host tissues ?
- 7) As suggested in point 5 above, if the protein levels for Mettl16 shown in Fig. 2G are significantly different between pairs of comparisons across mice, it would be helpful to provide a venn diagram showing the overlap between Mettl16 direct targets and the differentially methylated gene set observed in this study, in support that Mettl16 is the significant contributor for differential methylation. Publicly available datasets providing the direct (or overexpression/knockdown) targets of METTL16 in at least one tissue type could be used for such an analysis.
- 8) The abstract doesn't summarize several key observations such as functions affected, different motifs observed across tissues or differential m6A impact of the various bacteria studied or differential expression of mettl16 etc. but rather reads like a generic summary. It would help to see the major conclusions summarized quantitatively and qualitatively.

Minor concerns

Figure 2's legend needs to be corrected for grammatical consistency. Likewise, the manuscript's English is a bit rough in a few places although most of it reads great.

Pertaining to the "m6A/m modification profiles in cecum and liver": Do the methylation locations observed in this section are conserved across organisms (at least humans)? If these locations were observed to be methylated in phenotypes? or are these induced or novel methylation sites.

Page 4 line 124: How stringent was the DTU analysis? It is commonly seen that samples from same tissue sometimes have a different gene signature, especially at isoform level. However, in this DTU analysis section, there are only 6 genes reported have differential isoform usage, which is very uncommon, more details on this would help understand the isoform usage of tissues better.

Page 5 line 165: sentence reconstruction suggested.

Page 7 line 223: The premise of this study is very causal relationship dependent, it is assuming that microbiota is the only change in compared systems, that is inducing modifications. It is reasonable that understanding the mechanistic action behind this is beyond the scope of the study, but the analysis around Mettl16 is intriguing and would be interesting to see why only methylation writers were picked up and not methylation erasers.

Page 8 line 264: was Mat2a hypermethylated in these systems of mice too? Can a similar pattern be expected here (referring back to Mettl16 -Mat2a section)?

Reviewer #2 (Remarks to the Author):

Jabs et al. investigated the role of the gut microbiome in epitranscriptomic modifications of N6-methylation. While the overall study is interesting, some major concerns with regards to study design and analysis need to be addressed.

- Microbiome analysis: While the 16S analysis strategy sounds fine, the final number of OTUs mentioned in the methods section does not match the number of OTUs indicated on shaman.pasteur.fr. It is also not clear what was done with OTU classification at higher taxonomic levels and OTUs which could not be classified. If these OTUs were filtered and the OTUs re-normalized only including genus-level classifications (and species-level classification summarized to genus level), then this would substantially skew the abundances and potentially impact the results.
- Furthermore, the OTU table should be made available. I wasn't able to find it in any of the supplementary excel files and it didn't seem available on the website mentioned in the methods text. Please clearly indicate where the final OTU table can be found.
- Please motivate why the cecum and liver were tested rather than the colon, ileum, etc.
- What sample types were used to profile the gut microbiome? Stool?
- Please show the variation in microbiota composition. Does Figure S1 indicate averages over all samples? Variation in sample composition should be indicated and taken in to account. Why do the CONV profiles differ in S1B, C and D?
- How do the small number of mice in the mono-colonization experiments (e.g. 2 mice for E. coli) impact the number of m6A/m modifications that were observed?
- L 160: "this data suggests that the persisting commensals in vanco mice are involved in the gut microbiota-mediation regulation of m6Am medication patterns" -> It would be interesting to test mono-colonization of bacterial representatives of the depleted genera as control experiments. Are the modification patterns observed in connection with mono-colonization using any bacterium?

- L. 176-179: Please repeat the failed mono-colonization for E. coli.
- L 184-187: How as the effect size quantified? Were the different sample numbers taken into account?
- Additional controls: The effect of antibiotics on germ-free mice should be added as an additional control to test for changes due to antibiotic treatment itself.
- In addition to the pathway analysis, it would be interesting to discuss some of the top transcripts that were affected.
- How much is the transcript analysis affected by the varying and small number of samples, in particular for the mono-colonization experiments?
- For the Mettl16 and Mat2a analysis: Isn't it counterintuitive that the mono-colonized mice show comparable patterns to the GF mice?
- Ordination plots can visualize difference, however, the absence of a separation on the plot does not imply that there is no difference between the samples. Additional analysis is needed to support the statements (e.g. statements about the absence of group differences in reference to figure 1E and line 268/269 with regards to "a small effect").
- Parts of Figure 3 and 4 could be moved to the supplement. The text describing these figures is comparably short (~1/2 page for figure 3 and 4, respectively).
- The last paragraph of the paper should be more extensive, summarizing the findings and impact of the paper.

Figures:

- 1F+G: too small to clearly see patterns plus add axis labels for 1F
- 2A: what is measured here? Is it really the p-value on the xaxis? Why were "diseases and disorders" pathways not included in C and D?
- 2D: Why do all bars have the same length?
- 2E: Was this one gel? (Otherwise not comparable.) If so, please indicate in the legend.

Minor comments:

- Quantification and statistical analysis section: Please confirm that the cutoff mentioned in the last sentence refers to BH adjusted p-values and not the p-values themselves?
- Please specify the "minor modifications" in the LC-HRMS protocol.

Reviewers' comments:

Reviewer #1 (Remarks to the Author):

In this study, Jabs et. al present evidence for association between microbiota and host tissue's m6A RNA methylation. Extensive data resulting from multiple MeRIP experiments in cecum and liver across different mice models provides strong support for several claims in the study. However, there are several major concerns as summarized below making various aspects of study weak.

Major concerns

1) Manuscript title is very generic and reads like a review. Since the authors only focused on two tissues indicating different host tissues is not appropriate and justified. At the least the authors should extend the work to additional tissues or change the title to something more specific such as 'Impact of gut microbiota in on the m6A epitranscriptome in mouse cecum and liver tissues.'
We thank the reviewer for his criticism and have changed the title to 'Impact of gut microbiota on the m6A epitranscriptome in mouse cecum and liver tissues.'

2) In fact, it is unclear why the authors only limited to cecum and liver, the authors should at least provide the expression levels of methyltransferases across multiple mice tissues including brain to demonstrate a rationale for choosing these tissue types. For instance, it is increasingly known that immune tissues exhibit RNA methylation so including them to justify the choice would be appropriate.

Our choice of tissue was based on known influences of the microbiota on these tissues. The cecum is in direct contact with the microbiota which induces strong physiological and morphological changes in this organ. The liver is more remote, but nevertheless known to be strongly influenced by the microbiota. In addition, as stated in material and methods, MeRIPseq requires a large amount of input RNA (200ug for total RNA and about 5ug for mRNA or ribodepleted RNAs), so we chose tissues that would yield sufficient quantities of RNA from individual mice. The numbers of different populations of immune cells in germ-free mice are substantially reduced and a quantitative analysis might be quite difficult to perform. Nevertheless, we had collected additional organs (small intestine, colon and brain) from our different mouse models, and have performed Western Blotting and qRT-PCR to analyse the expression levels of methyltransferases and demethylases in these tissues in order to compare them with liver and cecum. We did not collect the spleen in all conditions, but we have compared expression levels of readers and erasers in the spleen with expression levels in the other organs for CONV mice as well. These data have been included in Figure S1.

3) The notion that MeRIP-seq also identifies m6Am is not well established and is still debatable. In addition, the authors neither show data or prior publication in support of this claim although this may indeed be the case in several loci for MeRIP-seq. So the authors should either cite a paper showing this evidence especially since m6Am positions have not been mapped accurately in mice tissues or provide data in support of this claim? In the absence of this, the authors should change throughout the manuscript m6A/m to m6A.

We thank the reviewer for his comment. Indeed, there are very few studies describing the positions of m6Am. The rationale behind referring to the detected peaks as m6A/m6Am is that the antibody used for precipitation does not distinguish between m6A and m6Am. However, as we have seen in LC-HRMS and also stated in the manuscript, m6Am is far less abundant than

m6A and located to very specific sites close to the cap-structure, which is why most of our detected peaks are likely m6A peaks. Thus, we have cited the corresponding studies and changed m6A/ m6Am back to m6A in the revised version of the manuscript. Additionally, we have analysed the mRNA expression for the recently discovered m6Am methyltransferase PCIF1 to compare its expression level in different tissues and to analyse if its expression is influenced by the gut flora. These data have been included in Figure S1A.

4) To show the reproducibility of the results across species and experimental systems, at the least the authors should demonstrate in human cell lines such as HepG2 (liver) in the presence of various commensal bacteria that m6A peaks are detected and provide a comparison of the overlap. Likewise, providing protein expression levels of Mettl's in human cell lines (with and without exposure to microbiota) should complement the claims.

Most of the commensals are anaerobes and thus cannot be co-cultured with cell lines. Consequently, the effects of a complex gut flora can only be incompletely studied *in vitro*. In addition, our results are obtained in a complex tissue composed of multiple different cell types. It is therefore unlikely that the effects we obtained in the different mouse models can be recapitulated in the proposed experiment. Nevertheless, this would be an interesting experiment to do for a follow-up study.

In Figure 1 we show the overlap of our detected m6A-peaks with peaks published in the MetDB-v2 database. We now have separated the calculation of the overlap to different tissues and cell lines of mouse and human origin and included this in FigureS2A. However, most studies have been focusing on cell lines and for most tissues the available datasets are rather small (e.g. one for mouse liver), so the conclusion drawn from these is limited. We would like to stress here again, that this is the first study comparing m6A-profiles between such a large number of conditions and complex tissues.

5) The authors present data in Fig 2E that the METL16 enzyme (methyltransferase) expression levels are different between CONV, mono-, germ-free mice and those in close contact with gut microbiota. However, no statistical data is shown, so it is unclear whether the trends are significant. Authors should present this data for all the METTLs analyzed for protein levels in both cecum and liver across the mouse models. In fact, to this reviewer it makes more sense to see this plot as one of the figure panels in Figure 1 before delving into MeRIP-seq data/results. In the absence of such differences in levels at RNA or protein, it is hard for this reviewer to know whether the differences are due to differential detection across samples or due to microbiota. We performed quantification of Mettl16 protein levels in the cecum, which confirmed statistically significantly reduced levels in GF and abx mice compared to CONV mice. These data have been included in Figure 4. Western Blot analyses and statistical quantification of additional methyltransferase (Mettl3, Mettl14) and the demethylase Alkbh5 levels for the other biological conditions in the cecum have been performed and were included in Figure S4 of the revised version of the manuscript. In addition, we compared and quantified the influence of a CONV flora on expression levels of Mettl3, 14, 16 and Alkbh5 in brain, small intestine, and colon on protein and mRNA level. We did not include the expression data in Figure 1 of the manuscript, since we are not convinced that the differential methylation is solely caused by slightly changed expression levels of the reader or writer proteins.

6) Would the data in Fig 1G still be the same if proteomic profiling is performed in couple of samples i.e, does differential methylation lead to differential protein levels in the host tissues? The correlation of RNAseq and proteomic data is known to be weak (e.g. Wang et al 2019). Nevertheless, we obtained proteomic data from CONV and GF cecum and compared them with m6A profiles and mRNA expression levels. These interesting data have been included in the revised version of the manuscript as Figure 2B. The mass spectrometry proteomics data have

been deposited to the ProteomeXchange Consortium via the PRIDE partner repository with the dataset identifier PXD016099. Reviewer account details:
Username: reviewer29981@ebi.ac.uk
Password: U3SU1Vxi

7) As suggested in point 5 above, if the protein levels for Mettl16 shown in Fig. 2G are significantly different between pairs of comparisons across mice, it would be helpful to provide a venn diagram showing the overlap between Mettl16 direct targets and the differentially methylated gene set observed in this study, in support that Mettl16 is the significant contributor for differential methylation. Publicly available datasets providing the direct (or overexpression/knockdown) targets of METTL16 in at least one tissue type could be used for such an analysis.

As stated above, Mettl16 levels are indeed significantly lower in GF mice than in CONV mice. Mettl16-targets in mouse tissue have not been mapped to our knowledge. We correlated peaks differentially methylated in GF versus CONV mice with published targets of METTL16 (Warda et al 2017 ; Pendleton et al. 2017) identified in human cell lines. However, the potential mRNA targets in those two studies only show a minimal overlap as you can see in the Venn diagram shown in Figure S5. Nevertheless, some of our differentially methylated peaks are present in the single datasets and in the overlap between the two, most notably Mat2a . Although it is unlikely that all the differentially methylated peaks we detect are merely due to changed Mettl16 expression levels, we thank the reviewer for his suggestion to include this comparison in the revised version of the manuscript.

8) The abstract doesn't summarize several key observations such as functions affected, different motifs observed across tissues or differential m6A impact of the various bacteria studied or differential expression of mettl16 etc. but rather reads like a generic summary. It would help to see the major conclusions summarized quantitatively and qualitatively.
We thank the reviewer for his comment and have changed the abstract in the revised version of the manuscript.

Minor concerns

Figure 2's legend needs to be corrected for grammatical consistency. Likewise, the manuscript's English is a bit rough in a few places although most of it reads great.
We have corrected the legends and hope that the revised manuscript reads better.

Pertaining to the "m6A/m modification profiles in cecum and liver": Do the methylation locations observed in this section are conserved across organisms (at least humans)? If these locations were observed to be methylated in phenotypes? or are these induced or novel methylation sites.

In general, m6A peaks in different organisms, tissues and cell lines have been described to be highly conserved (Meyer et al. 2012 ; Dominissini et al 2012) and dynamic changes between biological conditions have only been described in the context of viral infection (Tan et al. 2018). In addition to the overlap of all peaks we detected in cecum and liver, we computed the overlap of only the differentially methylated peaks with the reference peaks in the MetDB database (Figure S2). Most of the differentially methylated peaks in cecum (87%) and the liver (81%) are m6A peaks that have been mapped before. Interestingly, we found that there is very little overlap between differentially methylated peaks in cecum and liver (52, peaks, i.e. 2% and 12% of all differentially methylated peaks in cecum and liver, respectively), indicating that microbiota-dependent m6A profiles are tissue-specific.

Page 4 line 124: How stringent was the DTU analysis? It is commonly seen that samples from same tissue sometimes have a different gene signature, especially at isoform level. However, in this DTU analysis section, there are only 6 genes reported have differential isoform usage, which is very uncommon, more details on this would help understand the isoform usage of tissues better.

The number of genes exhibiting DTU provided in Table S3 for each comparison within each tissue is quite variable. The criteria for our DTU analysis were very stringent, as we used p-values adjusted using stageR package to control for overall false discovery rates simultaneously gene and isoform levels. We selected DTU cases by requiring both gene-level and transcript-level adjusted p-values lower than 5% and a change in isoform fraction that was larger than 0.1. We chose the high level of stringency to lower the number of false positives that could be caused by a relatively low sequencing depth.

Page 5 line 165: sentence reconstruction suggested. We have changed the sentence.

Page 7 line 223: The premise of this study is very causal relationship dependent, it is assuming that microbiota is the only change in compared systems, that is inducing modifications. It is reasonable that understanding the mechanistic action behind this is beyond the scope of the study, but the analysis around Mettl16 is intriguing and would be interesting to see why only methylation writers were picked up and not methylation erasers.

We had initially included the demethylase Fto in our Western Blots and found it to be expressed in the cecum at very low levels. We have performed Western Blotting on the other tissues proposed above for the two known demethylases, Fto and Alkbh5. However, the detected level of Fto with the antibody we used (abcam ab92821) was low. As an alternative, we have analysed Fto expression levels by qRT-PCR and included the data in Figure S1.

Page 8 line 264: was Mat2a hypermethylated in these systems of mice too? Can a similar pattern be expected here (referring back to Mettl16 -Mat2a section)?

Mat2a is not expressed in the liver and the liver-specific isoform of the S-adenosyl methionine synthase, Mat1a, is not known to be methylated by Mettl16. Furthermore, Mat1a is not differentially methylated in a microbiota-dependent manner in our datasets. Thus, a similar pattern is not expected here and there must be additional mechanisms present in the liver.

Reviewer #2 (Remarks to the Author):

Jabs et al. investigated the role of the gut microbiome in epitranscriptomic modifications of N6-methylation. While the overall study is interesting, some major concerns with regards to study design and analysis need to be addressed.

- Microbiome analysis: While the 16S analysis strategy sounds fine, the final number of OTUs mentioned in the methods section does not match the number of OTUs indicated on shaman.pasteur.fr. It is also not clear what was done with OTU classification at higher taxonomic levels and OTUs which could not be classified. If these OTUs were filtered and the OTUs re-normalized only including genus-level classifications (and species-level classification summarized to genus level), then this would substantially skew the abundances and potentially impact the results.

We thank the reviewer 2 for bringing this point to our attention. In the original version of the manuscript, we had included the experiment displayed in Figure 1D and did not update the OTU numbers accordingly. These were two independent 16S sequencing experiments, the first study displayed in Fig. S1B and S1C (key 1a0e447618724) and the second displayed in Fig. S1D (key 1a0e447618723). This was changed in the revised version of the manuscript. We combined the

the data of the two different sequencing experiments for the revised version of the manuscript in the new Figure S3 (Key = ff9551570bf15).

We excluded reads matching mouse sequences, since these would have produced OTUs with no annotation against SILVA. Other than that, we did not perform any filtering or re-normalization of the OTUs and all the amplicons (merged reads) were included in the OTU clustering.

The normalization of the contingency table was performed in SHAMAN at the OTU level (as described in McMurdie and Holmes 2014). Normalized counts are summed at the genus level when the OTU had $\geq 94.5\%$ identity with a reference in SILVA. Genus annotations with an identity level below this threshold were removed. This criterion was described in Yarza et al. 2014.

- Furthermore, the OTU table should be made available. I wasn't able to find it in any of the supplementary excel files and it didn't seem available on the website mentioned in the methods text. Please clearly indicate where the final OTU table can be found.

We made the experiments (Key = ff9551570bf15) available on figshare <https://doi.org/10.6084/m9.figshare.8321165.v1>. They are also available on Shaman in section raw data, in the panel Download .zip file.

- Please motivate why the cecum and liver were tested rather than the colon, ileum, etc. See response to reviewer 1 (point2). We changed the text in this revised version to better explain our choice of tissue.

- What sample types were used to profile the gut microbiome? Stool?

We used cecal content for 16S DNA sequencing. We included the information in the legend of Fig. S3 and in the methods section.

- Please show the variation in microbiota composition. Does Figure S1 indicate averages over all samples? Variation in sample composition should be indicated and taken in to account. Why do the CONV profiles differ in S1B, C and D?

We performed several different experiments and accordingly had different CONV controls : Fig. 1B displays the colonisation experiment to generate ex-GF mice, so CONV mice here are different from CONV mice in 1C, in which the control mice were gavaged with the vehicle for the antibiotics and vancomycin solutions. Both experiments were performed three times within about 6 months and samples from all three experiments are represented in the bar graph. The experiment in Fig. 1D is a replicate of the antibiotics treatment experiment that was performed independently and roughly two years later. Slight changes in the microbiota due to the provider and cage effects are likely to account for the observed differences.

We now combined the data of the two different sequencing experiments for the revised version of the manuscript in the new Figure S3.

Additionally, we show the most abundant genera for the individual samples in Figure S3A, and the variation between samples represented as boxplots (Figure S3C) and principal coordinates analysis (Figure S3D).

- How do the small number of mice in the mono-colonization experiments (e.g. 2 mice for E. coli) impact the number of m6A/m modifications that were observed?

It is known that a small number of replicates increases the number of false positives when performing differential analyses. This has been extensively demonstrated in RNASeq analysis (see e.g. Sonesson and Delorenzi 2013). However, in our dataset, Ec (n=2) was not the condition with most differential peaks or transcripts, while the Lp condition (n=3) had equivalent or slightly more differential peaks or transcripts than the conditions with more replicates (see Table S3). We additionally added the number of peaks detected in each IP sample to Table S1.

Furthermore, we added comments in the results and discussion sections mentioning that some conditions had a small number of replicates and pointed out that it might affect the analyses.

Figure 1 for the reviewers

Number of peaks detected (at least 10 read counts) in at least two IP samples per condition

- L 160: “this data suggests that the persisting commensals in vanco mice are involved in the gut microbiota-mediation regulation of m6Am medication patterns” -> It would be interesting to test mono-colonization of bacterial representatives of the depleted genera as control experiments. Are the modification patterns observed in connection with mono-colonization using any bacterium?

We agree with the reviewer that this would be very interesting to test. However, including one new condition would have implied repeating all the other conditions too. Experimental design is absolutely crucial in RNA-seq experiments and it is very difficult to merge datasets from different independent experiments. The phenotype we detected was robust enough to compare the data from two independent studies, but only because substantial numbers (n=4) of the other biological conditions (CONV, GF, Am, abx, and vanco) were also included in the second experiment. We could not perform such a huge experiment again in a reasonable amount of time.

- L. 176-179: Please repeat the failed mono-colonization for *E. coli*.

We are aware that the contamination in *E.coli*-colonised mice and the small number of replicates for this condition were a problem. We have therefore excluded this condition, since it was not possible to perform a new MeRIPseq analysis for the reasons mentioned above.

- L 184-187: How as the effect size quantified? Were the different sample numbers taken into account?

In the statement « Analysis of differentially methylated peaks between GF mice and Am- (24), Lp- (640), and Ec- (26) mono-associated mice, respectively, revealed that among these three bacteria *L.plantarum* has the strongest effect on host RNA-methylation. », it is the number of differentially methylated peaks that we used to determine whether a condition was more affected than the other (-2 and 2 were used as fold changes thresholds for all analyses). We now mention in the discussion that some conditions like Lp and ex-GF had low number of samples and that it may affect the differential analyses.

- Additional controls: The effect of antibiotics on germ-free mice should be added as an additional control to test for changes due to antibiotic treatment itself.

The experiment had been performed. However, we did not include the data in the manuscript. This experiment was technically complicated to perform in isolators(why?), which is why we had to use isocages instead. The risk of contamination in isocages is higher than in isolators. During gavage of GF mice every 12h with vehicle, antibiotics and vancomycin solutions, we encountered the problem that the vehicle-treated GF mice were contaminated with unknown

bacteria (See Figure 2 for the reviewers). Since these were supposed to serve for controlling for batch effects during RNA isolation, mRNA preparation and IP, the integration of the data obtained from antibiotics-treated GF mice (Figure 1 for the reviewers) will not allow a clear conclusion.

If we had repeated this experiment, the same comment as for including an additional commensal or repeating the *E.coli* colonization (see above) would apply. We agree that this is an interesting and important control. However, we would like to point out that antibiotics treatment is involved in only 2 out of the 8 conditions examined here, and that the number of differentially methylated peaks between CONV and GF mice is comparable to the differences between CONV and abx mice.

- In addition to the pathway analysis, it would be interesting to discuss some of the top transcripts that were affected. We agree with the reviewer and have included more details in the results and discussion.

- How much is the transcript analysis affected by the varying and small number of samples, in particular for the mono-colonization experiments?

As we mentioned above it is known that a small number of replicates increases the number of false positives when performing differential analyses in RNASeq analysis. However, in our dataset, Ec (n=2) was not the condition with the most differential transcripts, while the Lp condition (n=3) displayed equivalent or slightly elevated numbers of differential transcripts than the conditions with more replicates (see Table S3). We have added a sentence in the discussion mentioning that some conditions had a small number of replicates and will point out that it might affect the analyses.

- For the Mettl16 and Mat2a analysis: Isn't it counterintuitive that the mono-colonized mice show comparable patterns to the GF mice?

We are not very surprised by this finding. First of all, monocolonisation with one of the three bacteria tested does restore the m6A peaks in Mat2a. Additionally, quantification of the Western Blots have shown that Mettl16 and Mat2a levels are only significantly lower in GF and abx mice compared to CONV mice, which is equally the case for Mat2a expression. These data were included in Figure 4.

- Ordination plots can visualize difference, however, the absence of a separation on the plot does not imply that there is no difference between the samples. Additional analysis is needed to support the statements (e.g. statements about the absence of group differences in reference to figure 1E and line 268/269 with regards to "a small effect").

We have mentioned the numbers of differentially methylated peaks to illustrate the size of the effect.

- Parts of Figure 3 and 4 could be moved to the supplement. The text describing these figures is comparably short (~1/2 page for figure 3 and 4, respectively).

We thank the reviewer for his suggestion and have re-arranged the figures.

- The last paragraph of the paper should be more extensive, summarizing the findings and impact of the paper. We agree with the reviewer and have include a more detailed discussion.

Figures:

- 1F+G: too small to clearly see patterns plus add axis labels for 1F

We have changed the figure accordingly and moved the data to Figure 2A.

- 2A: what is measured here? Is it really the p-value on the xaxis? Why were "diseases and disorders" pathways not included in C and D?

We have included diseases and disorders in the pathway analysis.

- 2D: Why do all bars have the same length?

IPA gives a range of p-values, of which the lowest value happened to be the same.

• 2E: Was this one gel? (Otherwise not comparable.) If so, please indicate in the legend.
Yes, this was indeed one gel. We additionally performed quantification of the Western Blots using at least three different immunoblots (see response for reviewer 1).

Minor comments:

• Quantification and statistical analysis section: Please confirm that the cutoff mentioned in the last sentence refers to BH adjusted p-values and not the p-values themselves?

We have clarified this in the text and the respective figure legends.

• Please specify the “minor modifications” in the LC-HRMS protocol.

The modification has been specified.

Figure 2 for the reviewers

A, The eight mouse models analyzed: CONV, conventionally raised mouse; GF, germ-free mouse; ex-GF, GF mouse colonized with the gut flora of CONV mice; abx, CONV mice whose gut flora has been depleted by antibiotics treatment; vanco, vancomycin/amphotericinB-treated mice; GF abx, GF mice treated with antibiotics; Am, *A.muciniphila*-mono-colonized mice; Lp, *L.plantarum*-mono-colonized mice, **B**, Multi-dimensional scaling (MDS) plot of the peak log₂ counts-per-million IP data of all differentially methylated peaks showing the positions of the samples in the space spanned by the first and second MDS dimensions. Samples are coloured with respect to

condition. **C**, Heat map of the peak log₂ counts-per-million IP data; hierarchical clustering was performed using euclidean distance and ward.D2 linkage ; **D**, m⁶A/m-peaks found to be differentially methylated compared to differential expression of the entire transcripts GF vs GF abx (GF mice treated with antibiotics). Differentially methylated sites that are also differentially expressed on transcript level are in red, differentially methylated sites that are unchanged on transcript level, are in blue. m⁶A sites that were not significantly changed are shown in grey. Cut-offs for differential expression are log fold change (FC) -1 to 1 and Benjamini-Hochberg-corrected p-values <0.05. **E**, PCR of the V2 region of 16S RNA gene in feces from vehicle-treated (control), vancomycin- treated (vanco) and antibiotics (abx) treated GF-mice. gDNA isolated from CONV mice served as positive control (+).

Reviewers' comments:

Reviewer #1 (Remarks to the Author):

The authors have now addressed most of the concerns of this reviewer and I support the publication of this timely study. However, the manuscript would significantly benefit if the authors discuss Figure S1's results in detail within the manuscript. In this context, it is especially important to discuss the trends seen for the upregulation of the m6A writers in the spleen tissue and explain the rationale for not choosing this tissue for follow up studies such as the lack of sufficient RNA for follow up experiments.

Sarath Janga, PhD

Reviewer #2 (Remarks to the Author):

Remaining concerns:

The question why all OTUs are classified at the genus level remains. Typically, only a fraction of OTUs can be classified at the genus level. Excluding reads that map to the mouse genome is fine although this should be the case for very few reads as this is a 16S experiment. I couldn't find the reference "Yarza et al. 2014" in the manuscript. I assume the authors are referring to the paper corresponding to PMID 25118885. Yarza et al. propose a cutoff threshold of 94.5% for genus level characterization – that is fine. But does that mean that the authors exclude all reads that map to OTUs that do not have a genus level classification? For example, if only 30% of the reads map to OTUs with a genus-level classification, are 70% of the reads ignored because they mapped to OTUs that had <94.5% identity with a reference in SILVA? Do the authors normalize based on reads that have a genus level classification or using all reads?

Following the figshare link the authors provided for the OTU table led to three files: target_constitution.csv, target_colonization.csv and contrasts_constitution.txt – None of these contain the OTU table. Downloading the zip file from SHAMAN gave me access to the OTU table – however this table was not normalized for library size and contained OTUs without a genus level classification. This does not seem to be the OTU table that the authors used for their analysis. The authors should make the final OTU table available that was used to produce the results in this paper. An excel file as part of the supplement should work well.

Even if the monocolonization experiments and only-antibiotics are not feasible experiments to add, these limitations should be discussed in the paper.

Reviewer #1 (Remarks to the Author):

The authors have now addressed most of the concerns of this reviewer and I support the publication of this timely study. However, the manuscript would significantly benefit if the authors discuss Figure S1's results in detail within the manuscript. In this context, it is especially important to discuss the trends seen for the upregulation of the m6A writers in the spleen tissue and explain the rationale for not choosing this tissue for follow up studies such as the lack of sufficient RNA for follow up experiments.

We are pleased that the reviewer feels that his concerns have been adequately addressed. We have included an additional paragraph in the discussion explaining why we did not use the spleen or other immune cells for our analysis.

Reviewer #2 (Remarks to the Author):

Remaining concerns:

The question why all OTUs are classified at the genus level remains. Typically, only a fraction of OTUs can be classified at the genus level. Excluding reads that map to the mouse genome is fine although this should be the case for very few reads as this is a 16S experiment. I couldn't find the reference "Yarza et al. 2014" in the manuscript. I assume the authors are referring to the paper corresponding to PMID 25118885. Yarza et al. propose a cutoff threshold of 94.5% for genus level characterization – that is fine. But does that mean that the authors exclude all reads that map to OTUs that do not have a genus level classification? For example, if only 30% of the reads map to OTUs with a genus-level classification, are 70% of the reads ignored because they mapped to OTUs that had <94.5% identity with a reference in SILVA? Do the authors normalize based on reads that have a genus level classification or using all reads?

Following the figshare link the authors provided for the OTU table led to three files: target_constitution.csv, target_colonization.csv and contrasts_constitution.txt – None of these contain the OTU table. Downloading the zip file from SHAMAN gave me access to the OTU table – however this table was not normalized for library size and contained OTUs without a genus level classification. This does not seem to be the OTU table that the authors used for their analysis. The authors should make the final OTU table available that was used to produce the results in this paper. An excel file as part of the supplement should work well.

We apologize for the imprecisions that were still persisting in the description of the methods used for 16S analysis. We have rephrased the corresponding paragraph and provided more details on the numbers of mapped OTUs. Furthermore, we included normalized count tables for genus and OTU levels in Table S6. Additionally, we have uploaded the pre-print of the detailed description of SHAMAN and included the reference in the manuscript. We thank the reviewer for bringing to our attention that the reference Yarza et al. was lacking. It has been introduced in the text.

Even if the monocolonization experiments and only-antibiotics are not feasible experiments to add, these limitations should be discussed in the paper.

Monocolonization with *Akkermansia muciniphila* and *Lactobacillus plantarum* have been performed and discussed in detail. In contrast, the only antibiotics experiments were not conclusive and this is clearly indicated and additionally discussed in the text.

REVIEWERS' COMMENTS:

Reviewer #2 (Remarks to the Author):

Thank you for providing the OTU table and more details on the data processing. As far as I understand it now, the normalization was done at the OTU level. Counts for OTUs that map to the same genus are then summed up and only genus level associations are taken into account. I'm not sure why all other OTUs are discarded - they might be harder to follow up in experiments but could still provide interesting insights. However, since >90% of the reads map to these genus-level OTUs one might argue that the majority of the population is taken into account.

I tried to understand the normalization method that was used as I think it is important that the results are reproducible. I'm not familiar with the "weighted non-null normalization" that the authors mention.

The paper that is cited in that context (reference 104 by Bailey et al. on inferring direct DNA binding from CHIP-seq) doesn't mention this method either unfortunately. After some google searching I found the Shaman paper which was made available on bioarxiv now. The Shaman paper cites a paper by Mihai Pop. Pop et al. explain how they normalized their OTU table but didn't call it "weighted non-zero normalization". They in turn cite another paper for the full details of the normalization methods. Enough information should be provided in the current manuscript to follow the methods that were used to obtain the presented results.

Thank you for providing the OTU table and more details on the data processing. As far as I understand it now, the normalization was done at the OTU level. Counts for OTUs that map to the same genus are then summed up and only genus level associations are taken into account.

I'm not sure why all other OTUs are discarded - they might be harder to follow up in experiments but could still provide interesting insights. However, since >90% of the reads map to these genus-level OTUs one might argue that the majority of the population is taken into account.

We agree with the reviewer that we only consider annotated OTUs, which, as the reviewer correctly stated, is the vast majority. We may have lost some information here, but we are not attempting to address a new biological question with this experiment. We are simply controlling that the routinely performed standard treatments to deplete gut bacteria or enrich select genera has properly worked in our hands.

I tried to understand the normalization method that was used as I think it is important that the results are reproducible. I'm not familiar with the "weighted non-null normalization" that the authors mention.

The paper that is cited in that context (reference 104 by Bailey et al. on inferring direct DNA binding from CHIP-seq) doesn't mention this method either unfortunately. After some google searching I found the Shaman paper which was made available on bioarxiv now. The Shaman paper cites a paper by Mihai Pop. Pop et al. explain how they normalized their OTU table but didn't call it "weighted non-zero normalization". They in turn cite another paper for the full details of the normalization methods. Enough information should be provided in the current manuscript to follow the methods that were used to obtain the presented results.

We apologize that we forgot to update reference 104 from a previous version of the manuscript. We have corrected this mistake and cite reference 106, the preprint of the description of the SHAMAN tool.

We are, however, slightly puzzled that the reviewer had problems to find the preprint on bioRxiv, since it is appropriately cited and referenced with the correct doi in the manuscript several times. We would like to point out that Pop et al. is not cited in the context of the description of the normalization method in reference 106.

The weighted non-null normalization is described in subsection Normalization of the Description section (<https://doi.org/10.1101/2019.12.18.880773>).

It is an improvement of original relative log expression (RLE) normalization implemented in DESeq2.

In practice, the RLE method may lead to a defective normalization when only a few OTUs occur in all samples. In the weighted non-null normalization, cells with null values are excluded from the computation of the geometric mean. This method therefore takes all OTUs into account when estimating the size factor. Furthermore, weights are introduced so that OTUs with a big number of occurrences have a higher influence when calculating the

geometric mean. The impact of this new normalization technique was assessed in supplementary figure 2 of the manuscript by Volant. et al (reference 106 in our manuscript; see supplementary materials).

In addition, we provided a more detailed description of the normalization in the revised version of the manuscript.